# Insights into AMS/PCAT transporters from biochemical and structural characterization of a double Glycine motif protease

Silvia C Bobeica[1†], Shi-Hui Dong[2†], Liujie Huo[1‡], Nuria Mazo[3], Martin I McLaughlin[1], Gonzalo Jiménez-Osés[3,4], Satish K Nair[2,5*], Wilfred A van der Donk[1,2,6*]

[1]Roger Adams Laboratory, Department of Chemistry, University of Illinois at Urbana-Champaign, Urbana, United States; [2]Roger Adams Laboratory, Department of Biochemistry, University of llinois at Urbana-Champaign, Urbana, United States; [3]Departamento de Química, Centro de Investigación en Síntesis Química, Universidad de La Rioja, La Rioja, Spain; [4]CICbioGUNE, Derio, Spain; [5]Center for Biophysics and Computational Biology, University of Illinois at Urbana-Champaign, Urbana, United States; [6]Howard Hughes Medical Institute, University of Illinois at Urbana-Champaign, Urbana, United States

**\*For correspondence:**
snair@illinois.edu (SKN);
vddonk@illinois.edu (WAV)

[†]These authors contributed equally to this work

**Present address:** [‡]Shandong University-Helmholtz Institute of Biotechnology, State Key Laboratory of Microbial Technology, School of Life Sciences, Shandong University, Qingdao, China

**Abstract** The secretion of peptides and proteins is essential for survival and ecological adaptation of bacteria. Dual-functional ATP-binding cassette transporters export antimicrobial or quorum signaling peptides in Gram-positive bacteria. Their substrates contain a leader sequence that is excised by an N-terminal peptidase C39 domain at a double Gly motif. We characterized the protease domain (LahT150) of a transporter from a lanthipeptide biosynthetic operon in Lachnospiraceae and demonstrate that this protease can remove the leader peptide from a diverse set of peptides. The 2.0 Å resolution crystal structure of the protease domain in complex with a covalently bound leader peptide demonstrates the basis for substrate recognition across the entire class of such transporters. The structural data also provide a model for understanding the role of leader peptide recognition in the translocation cycle, and the function of degenerate, non-functional C39-like domains (CLD) in substrate recruitment in toxin exporters in Gram-negative bacteria.
DOI: https://doi.org/10.7554/eLife.42305.001

## Introduction

The translocation of peptides and proteins across the membrane bilayer is a fundamental process in all three domains of life (*Rapoport et al., 2017*). Bacteria secrete peptides and proteins for survival and adaptation to different ecological niches, to mediate intercellular signaling, and to deter or kill other microorganisms that may compete for limited resources (*Abele and Tampé, 2018*). Transport of intracellularly produced peptides in bacteria is mediated by ABC transporters, which utilize the energy of ATP binding and hydrolysis to translocate substrates across the bilayer (*ter Beek et al., 2014*; *Beis, 2015*). These transporters often contain an N-terminal protease domain, which belongs to the C39 class of cysteine proteases (Interpro:IPR005897) (*Rice et al., 2014*; *Michiels et al., 2001*). The dual-functional transporters, called ABC-transporter maturation and secretion (AMS) proteins or peptidase-containing ATP-binding transporters (PCAT), excise a leader sequence concomitant with

the unidirectional extracellular export of the mature peptide (*Håvarstein et al., 1995*; *van Belkum et al., 1997*).

In Gram-positive bacteria, the AMS/PCATs export peptides that mediate quorum signaling (*Pestova et al., 1996*) or exert antimicrobial activity upon removal of the leader peptide (*van Belkum et al., 1997*). Peptide substrates for the transporters contain leader sequences that typically end in a double Gly motif (Gly-Gly; Gly-Ala; Gly-Ser), and mutational analysis of the ComA transporter identified a consensus recognition sequence of Leu(−12)-(X)$_3$-Glu(−8)-Leu(−7) located N-terminal to this double Gly motif (*Ishii et al., 2010*).

In Gram-negative bacteria, the AMS/PCATs are a component of a membrane protein complex containing an outer membrane protein/factor (OMF) and an accessory protein to form the type I secretion system (T1SS) (*Létoffé et al., 1996*). Some T1SS complexes contain an AMS/PCAT transporter that lacks proteolytic activity due to the absence of the catalytically requisite Cys, and their N-terminal extension has thus been named C39-like domain (CLD) (*Kanonenberg et al., 2013*). Despite the lack of proteolytic activity, the CLD domain is necessary to recruit and tether the unfolded substrate during secretion through a yet-to-be established mechanism (*Lecher et al., 2012*).

The biosynthetic clusters of bacteriocins typically contain an AMS/PCAT exporter that directs the processing and secretion of double Gly-type leader sequences. Examples include colicin V and similar class II bacteriocins, and various classes of ribosomally synthesized and posttranslationally modified peptides (RiPPs) (*Arnison et al., 2013*). RiPPs are made from a precursor peptide that is morphed into a final bioactive product with a much larger structural diversity than can be achieved with the 20 proteinogenic amino acids. During this process, posttranslational modification enzymes often act iteratively on a subset of amino acids present in a C-terminal core peptide in a process that is directed by an N-terminal leader peptide (*Oman and van der Donk, 2010*). Following the installation of these posttranslational modifications, the cognate AMS/PCAT catalyzes the removal of the leader peptide via cleavage at the double Gly motif and extracellular secretion of the bioactive natural product (*Håvarstein et al., 1995*). Two particularly large classes of these double-Gly leader peptides belong to the Nif11 and nitrile hydratase families (*Haft et al., 2010*). Notably, the dual-functional transporters must ensure that modified and processed bacteriocins or RiPPs are directly shuttled out of the producing organism but the details for this fail-safe mechanism are not yet known.

The structure of a full-length PCAT from *Clostridium thermocellum* (termed PCAT1), containing both the C39 protease and the ABC transporter, demonstrated that the protease domain interacts with a transmembrane channel in the absence of bound nucleotide (*Lin et al., 2015*). Binding of ATP results in the disengagement of the protease domain from the helices of the transmembrane domain (TMD), suggesting an 'alternating-access' model for peptide translocation. Notably, biochemical studies of PCAT1 using a peptide substrate showed that proteolytic activity is enhanced during association with the TMD, but the underpinnings of this enhancement were unclear in the absence of a peptide substrate-bound structure.

Here, we identified and characterized LahT150, a sequence tolerant protease domain of the full-length AMS/PCAT LahT. We show that this protease can remove leader peptides from a large number of double Gly motif-containing peptides. In order to explain this tolerance for a broad range of substrates, we determined the 2.0 Å resolution crystal structure of the LahT protease domain in complex with a covalently bound leader peptide analog. Our structural and biochemical data provide insights into the determinants of substrate specificity. Modeling studies based on the structure of full-length PCAT1 provide insights how the AMS/PCAT transporters may prevent the escape of processed substrate into the cytoplasm of the producing cells. These studies explain much prior data on this large class of bifunctional exporters.

## Results and discussion

### Diversity and distribution of AMS/PCAT Transporters

We used the EFI-EST Tools (*Gerlt et al., 2015*) to generate a sequence similarity network (SSN) of likely AMS/PCAT members in GenBank. An alignment cutoff of at least 45% sequence identity was used to separate the clusters in this analysis. The results show that AMS/PCAT transporters are

distributed across various phyla of both Gram-positive and Gram-negative bacteria (*Figure 1*). Interestingly, the largest cluster from this SSN contains members from both Gram-positive (Firmicutes, Actinobacteria, Proteobacteria) and Gram-negative bacteria (Bacteroidetes, Cyanobacteria, Spirochaetes).

## Identification of a substrate tolerant protease

As a step toward advancing a biochemical and structure-function understanding of AMS/PCAT transporters and providing a tool for removing leader peptides from RiPP products, we sought to identify protease domains that retained catalytic activity in the absence of the TMD. Previous studies have demonstrated that the N-terminal 150 amino acids of AMS/PCAT proteins constitute peptidase C39 family members that can be expressed as individual active domains (*Håvarstein et al., 1995*; *Furgerson Ihnken et al., 2008*; *Ishii et al., 2006*; *Wu and Tai, 2004*; *Wang et al., 2016*). In search of a substrate tolerant protease domain, we first surveyed AMS transporters encoded in gene clusters containing multiple genes for precursor peptides with diverse core peptide sequences, because these proteases are expected to be inherently tolerant with respect to residues in the P' positions. We initially focused on the N-terminal protease domain of a transporter in *Prochlorococcus* MIT9313, which encodes 30 different RiPP substrates with leader peptides of the double Gly type (*Li et al., 2010*). Unfortunately, this domain did not prove active in our hands. We next turned to the protease domain of the FlvT transporter from *Ruminococcus flavefaciens* FD-1, encoded in a cluster with 12 substrate peptides (*Zhao and van der Donk, 2016*), but it also did not provide the desired robust activity. The lack of activity for these excised C39 protein domains is not unexpected, given prior studies that show that association with the TMD significantly enhances proteolytic activity of the protease domain in PCAT1 (*Lin et al., 2015*) and NukT (*Nishie et al., 2011*).

We next investigated an AMS/PCAT transporter LahT encoded in a gene cluster in a member of the human commensal microbiota, *Lachnospiraceae bacterium* C6A11 (genome from Bioproject ID 223496/Accession PRJNA223496). This cluster encodes nine different putative precursor peptides (LahA1-9) with diverse core peptide sequences and relatively conserved double Gly-motif leader peptides of the Nif11 type (*Figure 2*). Expression of the N-terminal 150 amino acids of LahT as a His$_6$-tagged fusion protein resulted in an active protease termed LahT150 that readily removed the leader peptide from the seven tested substrates encoded in the *lah* cluster at the predicted double Gly motif as monitored with matrix-assisted laser desorption time-of-flight mass spectrometry (MALDI-TOF MS) (*Figure 3* and *Figure 3—figure supplement 1*). Given the high divergence of the core peptide sequences of these peptides (*Figure 2*), the enzyme is highly tolerant to variation of the substrate sequence in the P' positions.

## Substrate tolerance of LahT150

Having established that LahT150 is highly forgiving with respect to the sequence of the core peptide of its cognate substrates as illustrated by the amino acids accepted after the cleavage site (*Figure 2* and *Figure 3*), we investigated its tolerance toward variations in the leader peptide. Select members of the ProcA peptides from *Prochlorococcus* MIT 9313 were first tested. This strain makes up to 30 different prochlorosins, members of the lanthipeptide family that are characterized by multiple thioether crosslinks introduced by posttranslational modifications (*Figure 4A*). Removal of their leader peptides has been challenging and has hampered production of the mature lanthipeptide products (*Tang and van der Donk, 2012*). We therefore tested LahT150 with a selection of ProcA peptides that were first posttranslationally modified by ProcM to introduce a variety of thioether ring structures (*Li et al., 2010*; *Shi et al., 2011*; *Bobeica and van der Donk, 2018*). The ProcM-modified ProcA peptides tested (five in total) proved to be substrates for LahT150 (*Figure 4B* and *Figure 4—figure supplement 1*), producing the mature prochlorosins (Pcns) and demonstrating that the enzyme can process substrates that contain polycyclic structures in the core peptide.

The ProcA and LahA leader peptides are relatively close in sequence (*Figure 4—figure supplement 2A*) and are all members of the Nif11 family. We next tested a series of maltose binding protein-tagged peptides that are phylogenetically more distant. Remarkably, LahT150 was able to cleave the leader peptide from a group of peptides encoded in *Azospirillum* sp. B510 (*Figure 4C* and *Figure 4—figure supplement 2B*) even though these peptides have leader peptides of the nitrile hydratase family (for sequences see *Figure 4—figure supplement 2A*).

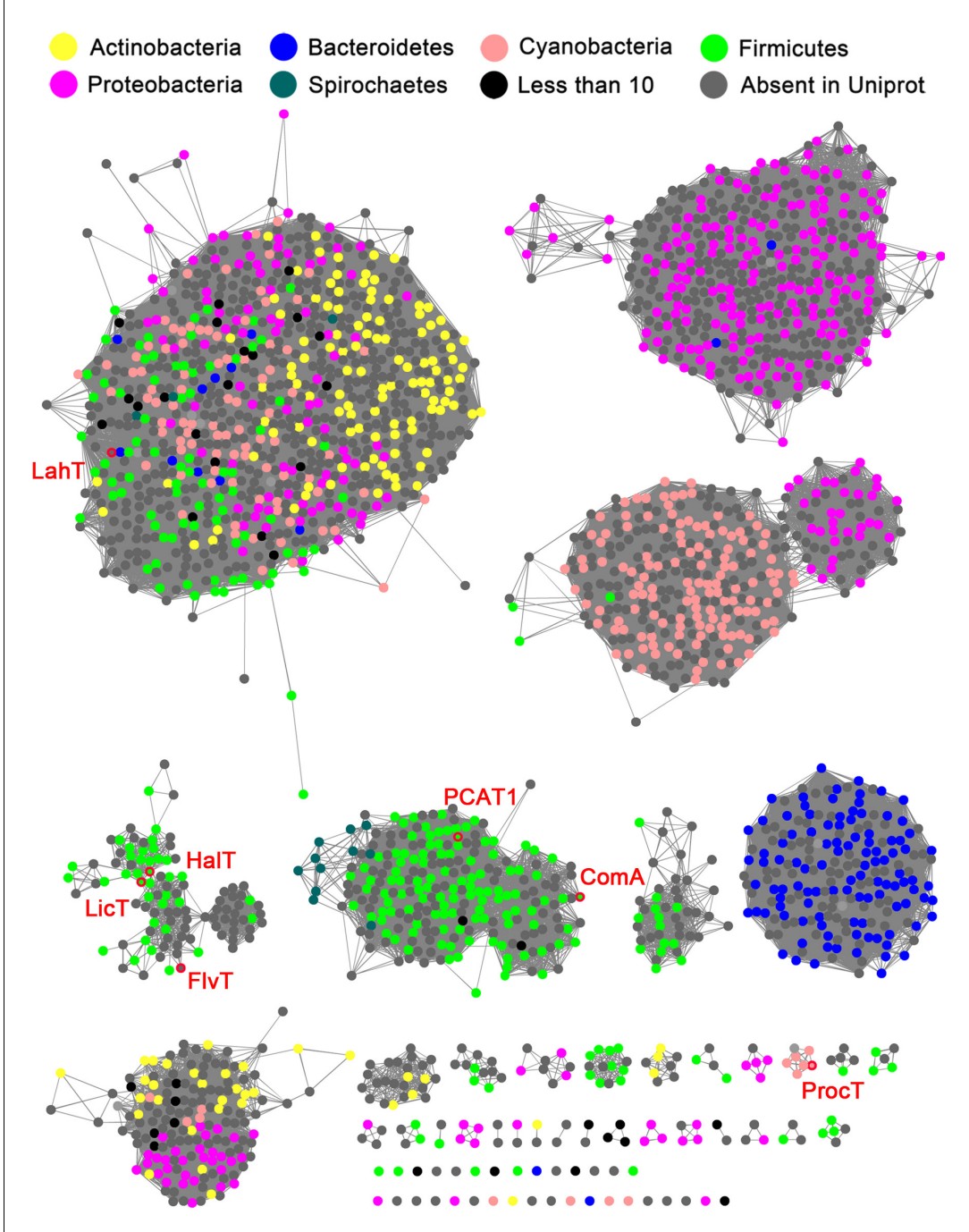

**Figure 1.** Sequence similarity network (SSN) of select full length AMS/PCAT proteins. Alignment cutoff of at least 45% sequence identity was applied to separate the clusters. The nodes representing LahT homolog sequences are colored by their corresponding phylum. Nodes of several characterized LahT homologs are marked by red circles and are labeled. The SSN tool draws sequences from UniProt; To increase the coverage of the network, additional sequences not in UniProt were added manually (grey nodes).

DOI: https://doi.org/10.7554/eLife.42305.002

## Determination of a minimum substrate recognition motif

As shown in the sequence alignment in *Figure 4—figure supplement 2A*, the peptides that are successfully cleaved after the double Gly motif are quite diverse in sequence, but some conservation can be detected. To determine a minimal substrate sequence, we first deleted the N-terminal 10,

20, and 30 residues from ProcA2.8. The resulting N-terminally truncated peptides were all substrates for LahT150 (*Figure 4—figure supplement 3A–E*). We then treated some of the substrates with commercial proteases to further trim the leader peptide. We first used a ProcA derivative (XY33a, *Figure 4—figure supplement 3A*) that expresses well and that was recently investigated in a screen for peptides that could inhibit the interaction between the UEV domain of TSG101 and the HIV p6 peptide (*Yang et al., 2018*). We digested XY33a with trypsin resulting in a truncated leader peptide containing only its C-terminal 14 residues attached to the core peptide (for sequence see *Figure 4—figure supplement 3A*). This truncant was a good substrate for LahT150 (*Figure 4—figure supplement 3E*). Next, we purchased a synthetic substrate encoding the last 13 amino acids of the leader peptide followed by Ala-Ala-Ser-Leu. This 17-amino acid peptide was cleaved by LahT150 at the expected position, which suggested that the recognition motif resides in the 13 C-terminal amino acids of the leader peptide (*Figure 4—figure supplement 3F*). Given this relatively short motif, we also investigated other RiPP precursor peptides that have much shorter leader peptides than the Nif11 and nitrile hydratase-type. The precursor peptides for haloduracin β and lacticin 481, lanthipeptides that have been previously produced in *E. coli* (*Shi et al., 2011*; *Oman et al., 2012*), were also successfully cleaved after GA and GS sequences even though the sequence homology is relatively low (*Figure 4D and E*; for sequences see *Figure 4—figure supplement 2A*). Cleavage after the GS sequence was also observed for the glycocin precursor peptide SunA (*Figure 4F*) (*Oman et al., 2011*), demonstrating the extension of the utility of LahT150 to a different RiPP class.

Site-directed mutagenesis was used next to provide information regarding critical residues. Positions −1 and −2 (the double Gly motif) have been investigated previously for other AMS enzymes (*Furgerson Ihnken et al., 2008*) and we therefore focused on residues N-terminal to this motif. Position −3 is almost invariably an Ala (*Figure 4G*) but variants of XY33a in which this Ala was mutated to Tyr, Phe, Lys, or Glu were all cleaved by LahT150 (*Figure 4—figure supplement 4*). Position −4 is usually a Val, Leu or Ile. Mutagenesis to Lys or Asp resulted in mutants that were not substrates for LahT150, but the XY33a V−4T mutant was completely processed. Surprisingly, the nearly invariant Glu at position −5 was not important as mutation to Lys, Ala, and Asp was tolerated (*Figure 4—figure supplement 4*). The same observation was made for the Glu at position −8, since mutation to Lys, Ala, and Asp did not abrogate catalysis. In contrast, mutation of Leu−7 to Lys or Asp had a strong negative effect on catalysis indicating that this residue is important (*Figure 4—figure supplement 4*). Mutagenesis of Leu−12 to Ala resulted in a peptide that was partially processed by LahT150; however, changing this residue to a Lys, Asp, Phe or Trp significantly impaired cleavage activity (*Figure 4—figure supplement 4*).

## Protease structure and substrate recognition

We next sought to obtain a structure of the protease domain with a covalently bound substrate analog that contained the leader sequence. LahT150 is a Cys protease, and substrate analogs terminating in an aldehyde at the scissile amide have been used successfully to obtain covalently bound inhibitors (*Westerik and Wolfenden, 1972*; *Thompson, 1973*). We therefore synthesized peptide **1** representing the C-terminal 13 amino acids of the ProcA2.8 leader peptide with an aldehyde in the

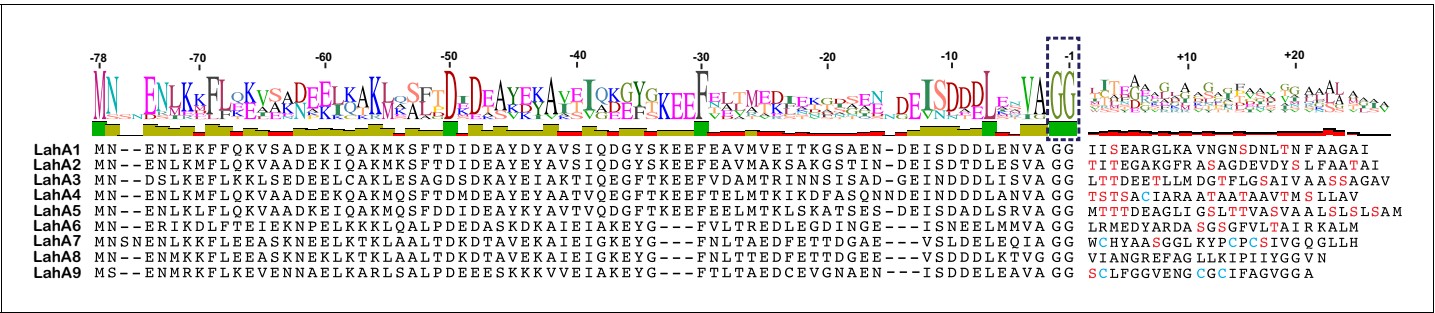

**Figure 2.** (A) Precursor peptides encoded in *Lachnospiraceae* C6A11 highlighting the conservation in the leader peptide with a sequence conservation logo (*Crooks et al., 2004*). The double Gly motif is boxed.
DOI: https://doi.org/10.7554/eLife.42305.003

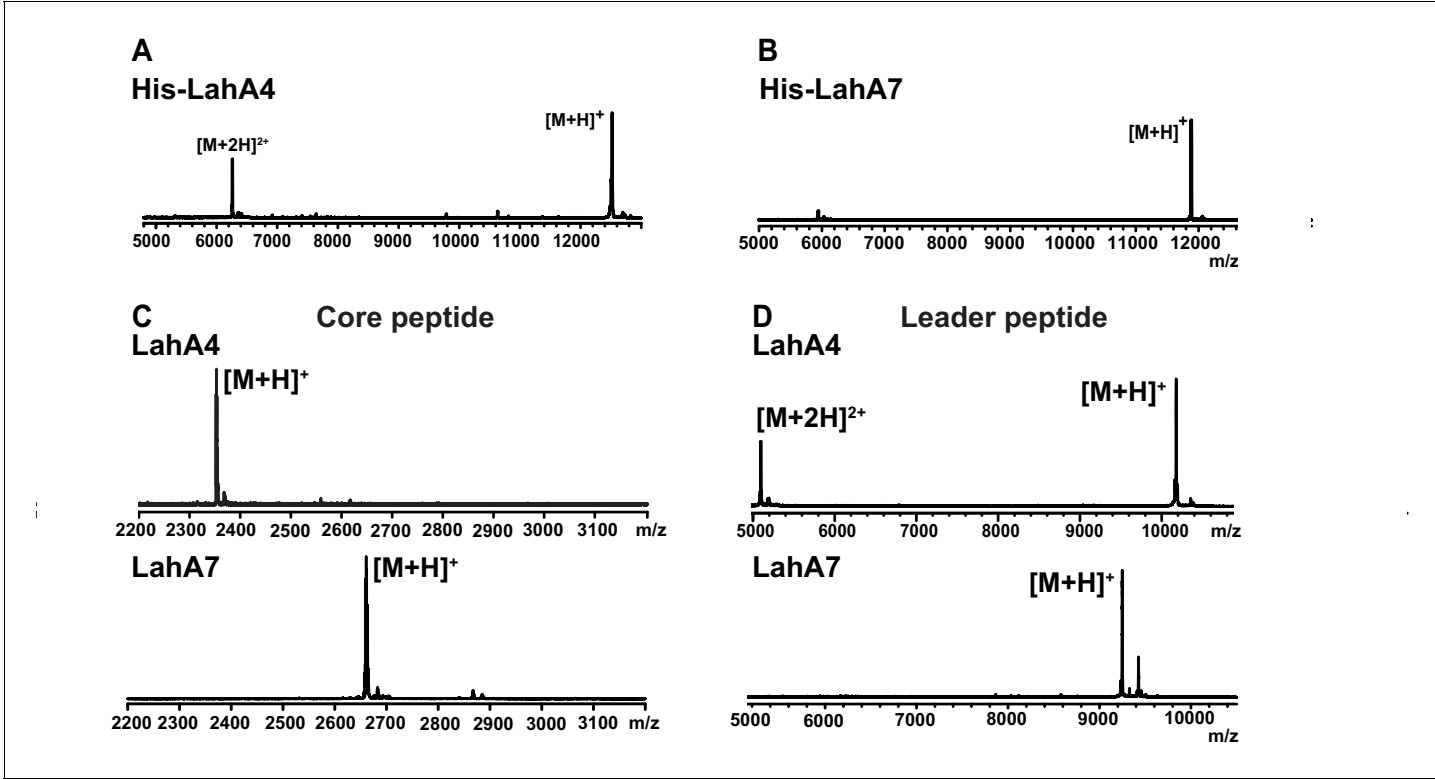

**Figure 3.** MALDI ToF MS analysis of proteolytic leader peptide removal of two LahA substrates catalyzed by LahT150. (A–B) MALDI ToF MS analysis of full length N-terminally hexahistidine tagged LahA4 and LahA7. (C) MALDI ToF MS analysis of the core peptides of LahA4 and LahA7 after LahT150 cleavage. Core peptide masses are [M + H]⁺: LahA4 (calcd 2352.2; obsd 2352.0), LahA7 (calcd 2660.3; obsd 2659.8). (D) MALDI TOF MS analysis of the leader peptides of LahA4 and LahA7 after LahT150 cleavage. Leader peptide average masses are [M + H]⁺: LahA4-leader (calcd 10188.8, obsd 10187.5), LahA7-leader (calcd 9246.9, obsd 9248.7). For five additional LahA substrates, see *Figure 3—figure supplement 1*.

DOI: https://doi.org/10.7554/eLife.42305.004

The following figure supplement is available for figure 3:

**Figure supplement 1.** LahT150 cleaves all LahA peptides.

DOI: https://doi.org/10.7554/eLife.42305.005

terminal position (*Figure 4H*). Crystallization efforts with the LahT150 expression construct failed, and we reasoned that the presence of flexible regions, including the His$_6$-tag may pose a hindrance. A shorter construct, including residues 1–147 (hereafter LahT147; as identified by secondary structure analysis) and a TEV cleavable His$_6$-tag, was used for crystallography. Incubation of peptide **1** with LahT147 prior to crystallization yielded crystals of the binary complex that diffracted to 2.0 Å resolution. Crystallographic phases were determined by the single wavelength anomalous diffraction method using data from a mercury-soaked crystal. The crystallographic asymmetric unit contains four copies of the complex allowing for multiple independent and unbiased views of the protein-peptide complex structure.

The overall structure of LahT147 recapitulates the α/β fold observed in the structures of other papain-like peptidase C39 members, wherein a central six-stranded antiparallel β-sheet is surrounded by five α-helices that pack on either side (*Figure 5A*). A DALI search against the Protein Data Bank shows that the closest structural homologs include the peptidase domain of the ComA transporter involved in quorum signaling (*Ishii et al., 2010*) (PDB Code 3K8U; Z-score = 21.1, RMSD of 1.3 Å over 129 aligned Cα atoms), and the protease domain of full-length PCAT1 *Lin et al., 2015* (PDB Code 4RY2; Z-score = 18.8, RMSD of 1.8 Å over 128 aligned Cα atoms). Notably, structural similarity is also detected with the non-functional C39-like domain (CLD) from the HlyB transporter from the T1SS involved in α-hemolysin secretion (*Lecher et al., 2012*) (PDB Code 3ZUA; Z-score = 14.5, RMSD of 2.4 Å over 123 aligned Cα atoms).

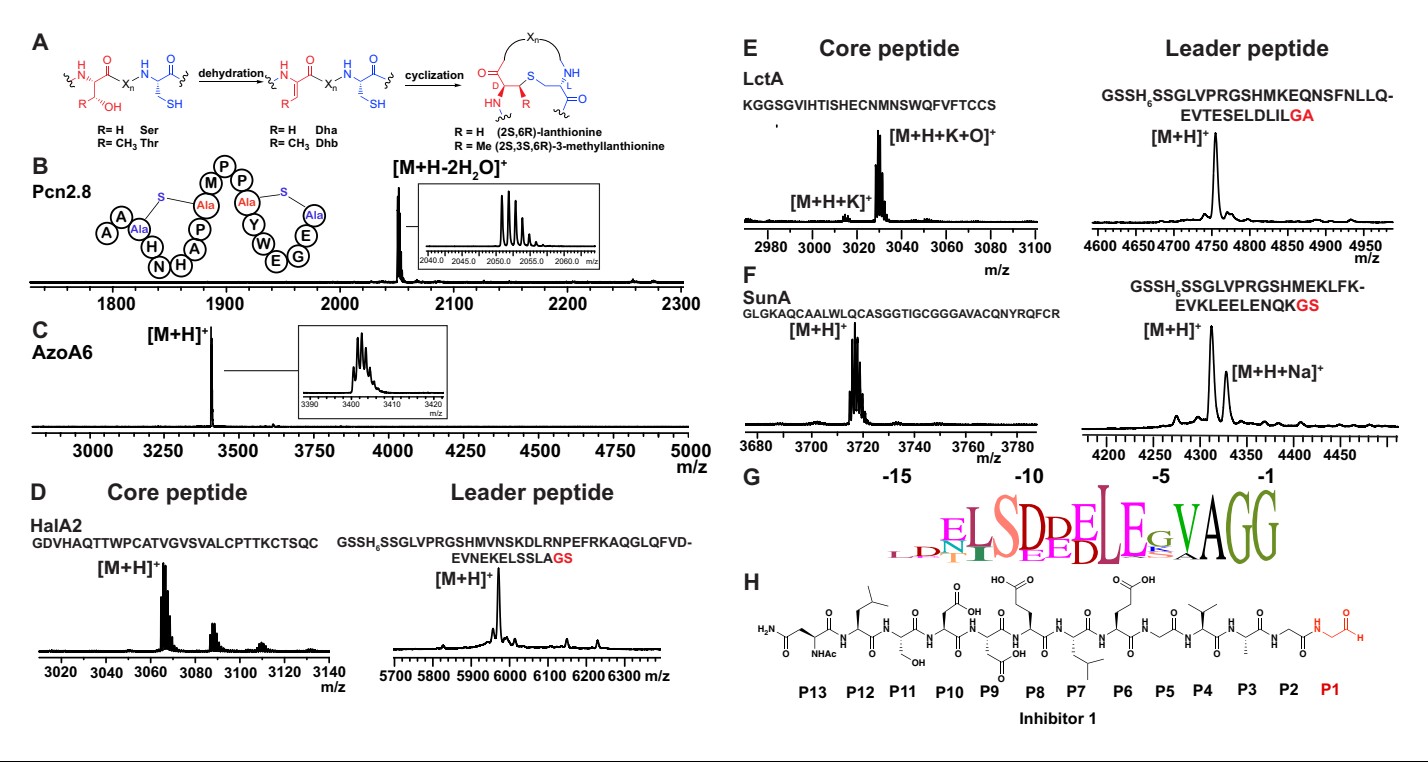

**Figure 4.** (A) Illustration of the posttranslational modifications in lanthipeptides. Serine and threonine residues are dehydrated by a lanthionine synthetase, resulting in dehydroalanine (Dha) and dehydrobutyrine (Dhb). The synthetase then catalyzes the Michael type addition of neighboring cysteine residues to the dehydrated residues. (B) Removal of the leader peptide of posttranslationally modified ProcA2.8 monitored by MALDI-TOF MS. Core peptide (two-fold dehydrated) [M + H]$^+$: calcd 2050.8, obsd 2050.9. For four additional ProcA substrates, see *Figure 4—figure supplement 1*. (C) In vitro leader peptide removal of AzoA6 bearing an N-terminal maltose binding protein tag. Core peptide [M + H]$^+$: calcd 3399.9, obsd 3400.4. For two additional AzoA substrates, see *Figure 4—figure supplement 2*. (D–F) MALDI TOF MS analysis of LahT150 catalyzed cleavage of the RiPP precursor peptides HalA2, LctA and SunA. Core peptide masses (left panels): HalA2 (calcd 3064.4; obsd 3064.6); LctA (calcd [M + H]$^+$ 3011.3 and [M + H + O]$^+$ 3027.3; obsd 3011.4 and 3027.4); SunA (calcd 3718.7; obsd 3718.6). Leader peptide ([M + H]$^+$) masses (right panels): HalA2-leader peptide (calcd avg. 5969.5; obsd 5969.5); LctA-leader peptide (calcd avg. 4754.2; obsd 4754.6); SunA-leader peptide (calcd avg. 4311.7, obsd 4311.2). (G) Sequence conservation logo (*Crooks et al., 2004*) showing the frequency of each amino acid (height of the letter) at the C-terminus of the 49 leader peptides in *Figure 4—figure supplement 2*. (H) Structure of peptide aldehyde inhibitor **1** based on the ProcA2.8 leader peptide.

DOI: https://doi.org/10.7554/eLife.42305.006

The following figure supplements are available for figure 4:

**Figure supplement 1.** Tests of LahT150 substrate tolerance with posttranslationally modified peptides.
DOI: https://doi.org/10.7554/eLife.42305.007

**Figure supplement 2.** Expanding LahT150 substrate tolerance to non-cognate peptides.
DOI: https://doi.org/10.7554/eLife.42305.008

**Figure supplement 3.** Determination of a minimum sequence for LahT150 catalysis.
DOI: https://doi.org/10.7554/eLife.42305.009

**Figure supplement 4.** Assessment of the importance of individual amino acids in the leader peptide for LahT150 catalysis.
DOI: https://doi.org/10.7554/eLife.42305.010

The catalytic triad of LahT147 is interspersed among the secondary structural elements, with His101 and Asp117 situated on strands β4 and β5, respectively, and the nucleophilic Cys27 on helix α1, which borders one side of the central sheet assembly. Clear and continuous density corresponding to the peptide aldehyde **1** can be observed bound covalently to Cys27 as a thiohemiacetal in all four crystallographically independent molecules. The peptide binds in a groove formed between helices α1–α3 and the central antiparallel β-sheet (*Figure 5A and B*). The C-terminus of peptide **1**, which consists of residues Gly−1 through Ala−3 that correspond to the P1 through P3 positions of the precursor substrate (per the nomenclature of Schechter and Berger [*Schechter and Berger, 1967*]) binds in a linear manner to the corresponding protease subsites. In contrast, residues Val−4

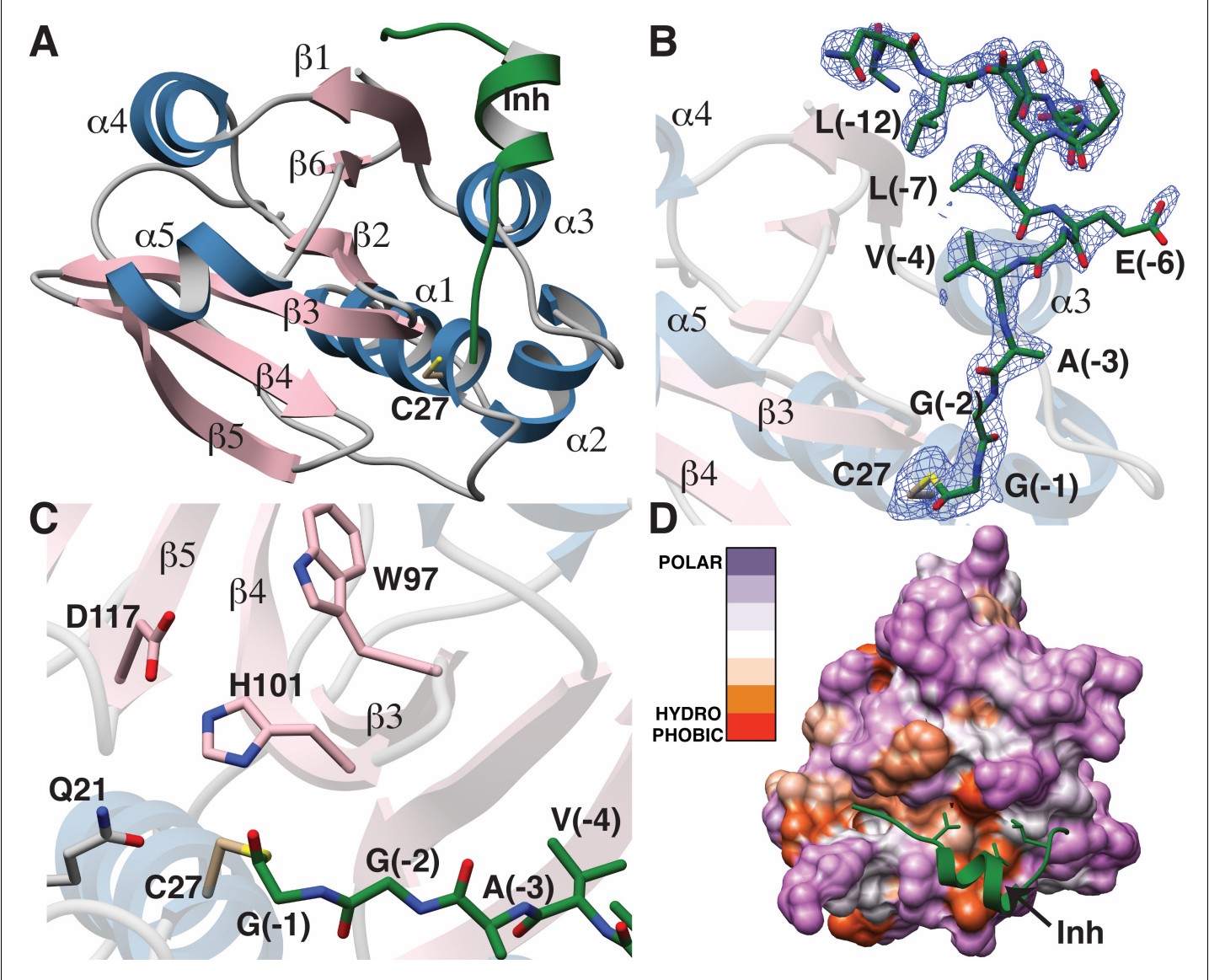

**Figure 5.** Structure of the LahT147-peptide aldehyde complex. (A) Overall structure of the complex showing the orientation of the peptide aldehyde (colored in green and labeled as Inh). (B) Simulated annealing difference Fourier map (calculated without the coordinates for Cys27 and the peptide aldehyde and shown at 2.3 σ) superimposed on the coordinates of the complex. (C) Close-up view of the active site showing residues implicated in catalysis. (D) Hydropathy analysis of LahT147 (based on the Kyte and Doolittle scale [*Kyte and Doolittle, 1982*]) superimposed in a color scheme onto a surface rendering of the final structure. Note that Val−4, Leu−7, and Leu−12 of the leader are positioned in suitable hydrophobic pockets. The figure was generated using the Chimera software package (*Pettersen et al., 2004*).

DOI: https://doi.org/10.7554/eLife.42305.011

through Ser−11 form a short two-turn helix, which positions residues Val−4, Leu−7, and Leu−12 of the leader peptide into a hydrophobic groove in LahT150, located roughly 20 Å away from the active site.

A closer view of the active site provides insights into the roles of specific residues in catalysis (*Figure 5C*). Prior studies on other AMS protease domains suggested that Gln21 may stabilize the oxyanion upon formation of the tetrahedral intermediate (*Ishii et al., 2010*), but in the cocrystal structure, this residue is located 4.3 Å away from the thiohemiacetal oxygen atom originating from the former carbonyl group. Another residue that can function in this role is the catalytic His101, which is positioned in line with the carbonyl oxygen at a distance of 2.7 Å. Studies of other

proteolytic enzymes, such as the PCY1 macrocyclase involved in orbitide biosynthesis, are also consistent with multiple functional roles for the catalytic His (*Chekan et al., 2017*).

Mapping the amino acid hydropathy onto a surface rendering of LahT147 (using the Kyte and Doolittle scale [*Kyte and Doolittle, 1982*] as implemented in the Chimera software package [*Pettersen et al., 2004*]) reveals that the surface of the protease largely consists of polar residues with the exception of the aforementioned hydrophobic groove that engages the two-turn helix of the substrate (*Figure 5D*). There is also an increase in hydrophobicity in the region flanking the active site, which is accompanied by a narrowing of the binding pocket. The hydrophobic packing interactions between LahT147 and peptide inhibitor **1** provide a rationale for our mutational data, wherein LahT147 could not process variants of the full-length precursor peptide that contained replacements of either Val−4 or Leu−7 with an acidic or basic residue nor most mutations of Leu−12 (*Figure 4—figure supplement 4*).

## Implications for proteolysis and transport

The structural and biochemical studies of an excised, active C39 protease afford the opportunity to understand prior data on substrate selectivity of AMS/PCAT transporters (*Michiels et al., 2001*). Our co-crystal structure illustrates that substrate recognition occurs roughly 20 Å away from the double Gly motif, and residues in this region form an α-helix that positions residues Val−4, Leu−7, and Leu−12 of the substrate peptide into a hydrophobic groove in the protease domain. Such 'knobs-into-holes' type packing is conceptually analogous to leader peptide recognition in RiPP biosynthetic enzymes (*Ortega et al., 2015*; *Koehnke et al., 2015*; *Grove et al., 2017*; *Evans et al., 2017*) wherein hydrophobic residues in the precursor peptide, distal from the core peptide are positioned into suitably arranged nonpolar pockets.

A plausible model for substrate translocation in the context of a full-length AMS/PCAT can be envisioned by superimposing the cocrystal structure of the LahT protease domain onto the structure of full-length PCAT1 in the absence of nucleotides (*Lin et al., 2015*). Despite overall low (~25%) sequence identity between the respective protease domains, the structures align with a RMSD of 1.6 Å over 128 Cα atoms. In the model, the helical region of the substrate peptide is oriented in a small groove located between the protease domain and the nucleotide-binding domain, providing an effective means to lock these domains together. The substrate peptide cargo would be directed out from the active site of the protease domain and positioned directly into the TMD to facilitate export. This pocket is increasingly narrow near the site of proteolysis, explaining the identity of small residues at the double Gly cleavage site (*Figure 6A*). There is also an increase in the hydrophobicity of the pocket near this site, which may serve to further direct the substrate into the TMD. The leader peptide may provide a backstop that prevents the cleaved cargo from leaking back into the cytoplasm, and ensuring the extracellular directionality of transport upon binding of ATP to the NBDs. The conformational changes observed in the structure of PCAT1 upon nucleotide binding both ensure shuttling of the cleaved cargo to the extracellular region, and result in disengagement of the protease domain for subsequent rounds of coupled export (*Lin et al., 2015*).

The structural and biochemical data presented here also inform on the cryptic non-catalytic C39-like domains (CLDs) associated with type I secretion systems in Gram-negative bacteria. Analysis of transport of hemolysin A (HlyA) mediated by the ABC transporter HlyB demonstrates that the CLD is essential for secretion, despite lacking proteolytic activity (*Lecher et al., 2012*). Pull-down assays demonstrate that the CLD interacts only with the unfolded HlyA substrate, leading to the suggestion that the CLD may play a chaperone-like role. Our studies suggest that the CLD may also play a role in stabilizing the nucleotide-free state of HlyB by binding the substrate peptide, which may optimally position the peptide cargo for export upon ATP binding.

## Utility of the LahT150 protease domain

Because of the direct link between the genome-encoded precursor peptide and the final natural product, RiPPs are attractive for genome mining (*Velásquez and van der Donk, 2011*; *Hetrick and van der Donk, 2017*) and synthetic biology (*Yang et al., 2018*; *Sardar et al., 2015*; *Burkhart et al., 2017*; *van Heel et al., 2013*; *Montalbán-López et al., 2017*; *Urban et al., 2017*; *Hetrick et al., 2018*), and numerous studies have described the successful reconstitution of RiPP machinery in heterologous hosts, including *E. coli*. In many such genome mining exercises, the leader peptide is not

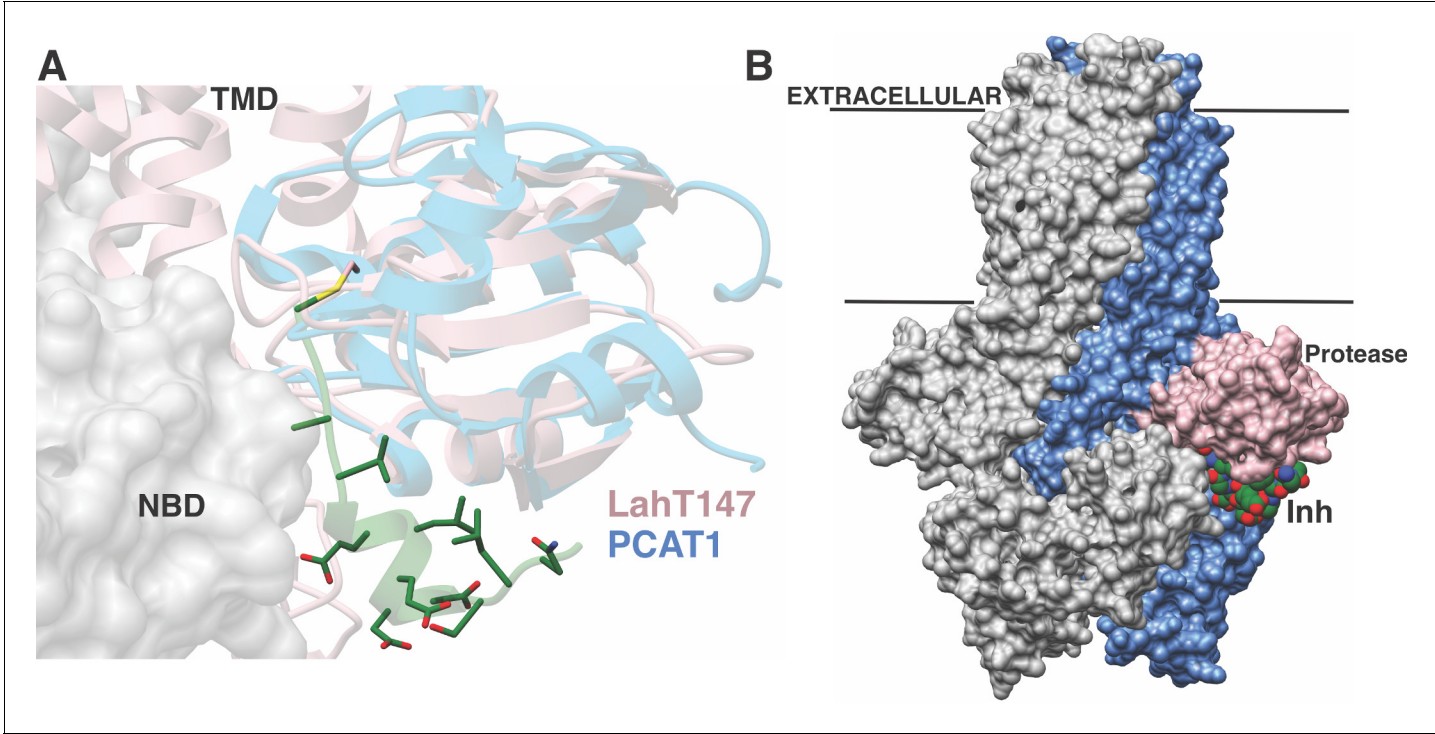

**Figure 6.** Structure-based superposition of LahT147 and PCAT1. (**A**) Close-up view of the LahT147-inhibitor complex structure superimposed on the crystal structure of full-length PCAT1. Note that the leader sequence directs the core peptide 'cargo' into the transmembrane domain (TMD) and is flanked by the nucleotide-binding domain (NBD). (**B**) Overall structure of the PCAT1 dimer with one monomer colored grey and the other monomer blue and pink showing the relative orientations of the protease domain and the inhibitor. Binding of the peptide cargo is poised to stabilize the interdomain interactions in the full-length transporter.

DOI: https://doi.org/10.7554/eLife.42305.012

removed inside the heterologous host to prevent potential toxicity of the final compound (*Valsesia et al., 2007*). An additional advantage of retaining the leader peptide during heterologous production is that it enables attachment of an affinity tag that allows one-step purification, which avoids tedious purification of compounds that may be produced in small quantities (*Shi et al., 2011*; *Nagao et al., 2005*). However, developing general methods for leader peptide removal has been challenging (*Li et al., 2010*; *Shi et al., 2011*; *Goto et al., 2010*; *Plat et al., 2011*; *Lohans et al., 2014*; *Ökesli et al., 2011*), especially for the double Gly leader peptides. This study shows that LahT150 is a highly versatile and useful protease for RiPP research as it removes the leader peptide from a remarkably diverse set of peptides including leader peptides of the Nif11 and nitrile hydratase families.

## Materials and methods

For all materials used or generated in this study, see the Key Resources Table.

**Key resources table**

| Reagent type (species) or resource | Designation | Source or reference | Identifiers | Additional information |
|---|---|---|---|---|

*Continued on next page*

*Continued*

| Reagent type (species) or resource | Designation | Source or reference | Identifiers | Additional information |
|---|---|---|---|---|
| Gene (XY33a_V-4K_gene) | XY33a_V-4K_gene | IDT. Representative of other purchased XY33a mutant genes (see *Table 5*) | XY33a_V-4K_gene | 5 ng/µL stock solution (1 µL) used as amplification template |
| Strain, strain background (*Escherichia coli* BL21 (DE3)-T1R) | E. coli BL21 (DE3)-T1R | Sigma Aldrich B2935 | BL21 (DE3)-T1R | |
| Strain, strain background (Lachnospiraceae C6A11) | Lachnospiraceae C6A11 | Dr. William Kelly (AgResearch, New Zealand) | Lachnospiraceae C6A11 | |
| Strain, strain background (*Escherichia coli* Rosetta 2 (DE3)) | *E. coli* Rosetta 2 (DE3) | Novagen Catalog no. 71400–3 | *E. coli* Rosetta 2 (DE3) | |
| Strain, strain background (Azospirillum sp. B510 (JCM 14679)) | Azospirillum sp. B510 (JCM 14679) | JCM Riken http://www.jcm .riken.jp/cgi-bin/jcm/jcm_ number?JCM=14679 | Azospirillum sp. B510 (JCM 14679) | |
| Transformed construct (pETDuet LahT150) | pETDuet-LahT150 | this work | pETDuet LahT150 | 50 ng/µL stock solution (1 µL) used in E. coli BL21 transformation |
| Transformed construct (pRSFDuet XY33a) | pRSFDuet-XY33a | PMID: 22574919 | pRSFDuet XY33a | 50 ng/µL stock solution (1 µL) used in E. coli BL21 transformation |
| Transformed construct (pRSFDuet XY33a A-3Y) | pRSFDuet XY33a A-3Y | this work. Representative XY33a mutant | pRSFDuet XY33a A-3Y | 50 ng/µL stock solution (1 µL) used in E. coli BL21 transformation |
| Transformed construct (pRSFDuet LahA1) | pRSFDuet-LahA1 | this work | pRSFDuet LahA1 | 50 ng/µL stock solution (1 µL) used in E. coli BL21 transformation |
| Transformed construct (pRSFDuet LahA2) | pRSFDuet-LahA2 | this work | pRSFDuet LahA2 | 50 ng/µL stock solution (1 µL) used in E. coli BL21 transformation |
| Transformed construct (pRSFDuet LahA3) | pRSFDuet-LahA3 | this work | pRSFDuet LahA3 | 50 ng/µL stock solution (1 µL) used in E. coli BL21 transformation |
| Transformed construct (pRSFDuet LahA4) | pRSFDuet-LahA4 | this work | pRSFDuet LahA4 | 50 ng/µL stock solution (1 µL) used in E. coli BL21 transformation |
| Transformed construct (pRSFDuet LahA5) | pRSFDuet-LahA5 | this work | pRSFDuet LahA5 | 50 ng/µL stock solution (1 µL) used in E. coli BL21 transformation |
| Transformed construct (pRSFDuet LahA6) | pRSFDuet-LahA6 | this work | pRSFDuet LahA6 | 50 ng/µL stock solution (1 µL) used in E. coli BL21 transformation |
| Transformed construct (pRSFDuet LahA7) | pRSFDuet-LahA7 | this work | pRSFDuet LahA7 | 50 ng/µL stock solution (1 µL) used in E. coli BL21 transformation |

*Continued on next page*

*Continued*

| Reagent type (species) or resource | Designation | Source or reference | Identifiers | Additional information |
|---|---|---|---|---|
| Transformed construct (pET28-MBP-AzoA2) | pET28-MBP-AzoA2 | this work | pET28-MBP-AzoA2 | 150 ng/μL stock solution (1 μL) used in E. coli Rosetta 2 (DE3) transformation |
| Transformed construct (pET28-MBP-AzoA3) | pET28-MBP-AzoA3 | this work | pET28-MBP-AzoA3 | 150 ng/μL stock solution (1 μL) used in E. coli Rosetta 2 (DE3) transformation |
| Transformed construct (pET28-MBP-AzoA6) | pET28-MBP-AzoA6 | this work | pET28-MBP-AzoA6 | 150 ng/μL stock solution (1 μL) used in E. coli Rosetta 2 (DE3) transformation |
| Transformed construct (pET28-MBP-AzoA7) | pET28-MBP-AzoA7 | this work | pET28-MBP-AzoA7 | 150 ng/μL stock solution (1 μL) used in E. coli Rosetta 2 (DE3) transformation |
| Transformed construct (pRSFDuet ProcA 2.8 (MCSI) - ProcM (MCSII)) | pRSFDuet ProcA2.8 (MCSI) - ProcM (MCSII) | PMID: 22574919 | pRSFDuet ProcA 2.8 (MCSI) - ProcM (MCSII) | 50 ng/μL stock solution (1 μL) used in E. coli BL21 transformation |
| Transformed construct (pRSFDuet ProcA 1.7 (MCSI) - ProcM (MCSII)) | pRSFDuet ProcA1.7 (MCSI) - ProcM (MCSII | PMID: 22574919 | pRSFDuet ProcA 1.7 (MCSI) - ProcM (MCSII | 50 ng/μL stock solution (1 μL) used in E. coli BL21 transformation |
| Transformed construct (pRSFDuet ProcA 2.1 (MCSI) - ProcM (MCSII)) | pRSFDuet ProcA2.1 (MCSI) - ProcM (MCSII | this work | pRSFDuet ProcA 2.1 (MCSI) - ProcM (MCSII | 50 ng/ μL stock solution (1 μL) used in E. coli BL21 transformation |
| Transformed construct (pRSFDuet ProcA 2.4 (MCSI) - ProcM (MCSII)) | pRSFDuet ProcA2.4 (MCSI) - ProcM (MCSII) | this work | pRSFDuet ProcA 2.4 (MCSI) - ProcM (MCSII) | 50 ng/ μL stock solution (1 μL) used in E. coli BL21 transformation |
| Transformed construct (pRSFDuet ProcA 1.3 (MCSI) - ProcM (MCSII)) | pRSFDuet ProcA1.3 (MCSI) - ProcM (MCSII) | this work | pRSFDuet ProcA 1.3 (MCSI) - ProcM (MCSII) | 50 ng/ μL stock solution (1 μL) used in E. coli BL21 transformation |
| Sequence-based reagent | Benzonase Endonuclease | EMD Millipore Catalog no. 1.01656.001 | Benzonase | |
| Sequence-based reagent | EcoRI-HF | New England Biolabs R3101S | EcoRI | |
| Sequence-based reagent | BamHI-HF | New England Biolabs R3136S | BamHI | |
| Sequence-based reagent | NotI-HF | New England Biolabs R3189S | Not1 | |
| Sequence-based reagent | HindIII-HF | New England Biolabs R3104S | HindIII | |
| Recombinant protein | XY33a | PMID: 29507389 | XY33a | recombinant substrate peptide tested with LahT150 |
| Recombinant protein | XY33a A-3Y | this work; representative XY33a mutant | XY33a A-3Y | recombinant substrate peptide mutant tested with LahT150 |
| Recombinant protein | LahA1 | this work | LahA1 | recombinant substrate peptide tested with LahT150 |

*Continued on next page*

*Continued*

| Reagent type (species) or resource | Designation | Source or reference | Identifiers | Additional information |
|---|---|---|---|---|
| Recombinant protein | LahA2 | this work | LahA2 | recombinant substrate peptide tested with LahT150 |
| Recombinant protein | LahA3 | this work | LahA3 | recombinant substrate peptide tested with LahT150 |
| Recombinant protein | LahA4 | this work | LahA4 | recombinant substrate peptide tested with LahT150 |
| Recombinant protein | LahA5 | this work | LahA5 | recombinant substrate peptide tested with LahT150 |
| Recombinant protein | LahA6 | this work | LahA6 | recombinant substrate peptide tested with LahT150 |
| Recombinant protein | LahA7 | this work | LahA7 | recombinant substrate peptide tested with LahT150 |
| Recombinant protein | AzoA2 | this work | MBP-AzoA2 | recombinant substrate peptide tested with LahT150 |
| Recombinant protein | AzoA3 | this work | MBP-AzoA3 | recombinant substrate peptide tested with LahT150 |
| Recombinant protein | AzoA6 | this work | MBP-AzoA6 | recombinant substrate peptide tested with LahT150 |
| Recombinant protein | AzoA7 | this work | MBP-AzoA7 | recombinant substrate peptide tested with LahT150 |
| Recombinant protein | LahT150 | this work | LahT150 | protease domain of LahT |
| Recombinant protein | Pcn 2.8 | PMID: 22574919 | Pcn 2.8 | recombinant posttranslationally modified ProcA2.8 substrate peptide tested with LahT150 |
| Recombinant protein | Pcn 1.7 | PMID: 22574919 | Pcn 1.7 | recombinant posttranslationally modified ProcA1.7 substrate peptide tested with LahT150 |
| Recombinant protein | Pcn 2.1 | this work | Pcn 2.1 | recombinant posttranslationally modified ProcA2.1 substrate peptide tested with LahT150 |

*Continued on next page*

*Continued*

| Reagent type (species) or resource | Designation | Source or reference | Identifiers | Additional information |
|---|---|---|---|---|
| Recombinant protein | Pcn 2.4 | this work | Pcn 2.4 | recombinant posttranslationally modified ProcA2.4 substrate peptide tested with LahT150 |
| Recombinant protein | Pcn 1.3 | this work | Pcn 1.3 | recombinant posttranslationally modified ProcA1.3 substrate peptide tested with LahT150 |
| Minimum peptide substrate | Synthetic peptide | Genscript | Synthetic peptide | synthetic minimal substrate peptide tested with LahT150 |
| Commercial kit | QIAprep Spin Miniprep kit | Qiagen catalog no. 27106 | QIAprep Spin Miniprep kit | |
| Commercial kit | QIAquick Gel Extraction kit | Qiagen catalog no. 28115 | QIAquick Gel Extraction kit | |
| Commercial kit | Gibson Assembly | New England Biolabs E2611S | Gibson Assembly | |
| Chemical compound | TCEP (Tris (2-Carboxyethyl) phosphine hydrochloride) | Goldbio Catalog ID TCEP | TCEP | |
| Chemical compound | Terrific Broth granulated | Fisher Scientific BP97285 | TB | |
| Chemical compound | Glycerol | Fisher Scientific BP-229–4 | Glycerol | |
| Chemical compound | Dextrose | Fisher Scientific BP350500 | Glucose or dextrose | |
| Chemical compound | kanamycin monosulfate, USP grade | Goldbio Catalog ID K-120 | kanamycin | |
| Software, algorithm | Adobe Illustrator CS6 | Adobe | Adobe Illustrator | |
| Software, algorithm | FlexAnalysis 3.4 (Bruker Daltonik GmbH) | Bruker Daltonik GmbH | FlexAnalysis 3.4 (Bruker Daltonik GmbH) | Mass spectrometry data processing |
| Other | Clontech His60 Ni Superflow resin | Clontech Catalog no. 636660 | Clontech His60 Ni Superflow resin | Used for gravity purification of all recombinant proteins except LahT150 |
| Other | GE Healthcare HisTrap HP | GE Healthcare 17524701 | 5 mL HiTrap Ni Chelating column | Used for FPLC purification of recombinant LahT150 |

*E. coli* BL21 (DE3)-T1[R] transformants containing pETDuet encoding N-terminally His-tagged LahT150 in MCSI were grown overnight at 37°C in Terrific Broth (TB) containing 2% glucose and 100 µg/mL ampicillin and used to inoculate production cultures. Cultures were grown aerobically at 37°C until $OD_{600}$ reached 0.8, then cooled on ice for 20 min before induction with 0.2 mM IPTG. The cells were then incubated at 18°C for 16–20 hr. Cells were harvested by centrifugation at 5,000 × $g$ at 4°C, then resuspended in LahT150 Lysis Buffer (20 mM Tris, 1 M NaCl, pH 7.8) and stored at −80°C until purification. To the thawed cells, 40 mg lysozyme and 10 µL benzonase (≥250 units/µL, EMD Millipore) per 12 g cell paste (approximate cell mass obtained from 1 L of culture) were added and the cells were incubated for 90 min in a beaker on ice. The cells were then sonicated at 60% amplitude for 6 min using 2 s pulse on and 8 s pulse off using a Sonics and Materials Inc. Vibra Cell VCX 500 or VCX 700 or passed through an Avestin C3 Cell Homogenizer. The lysed cells were then

centrifuged at 24,000 × *g* for 30 min at 4°C. The supernatants were transferred to new tubes and again centrifuged. The supernatant was applied to a 5 mL HiTrap Ni chelating column pre-equilibrated with 10 column volumes (CV) of LahT150 Lysis Buffer using a peristaltic pump using ~2 mL/min flow rate. The HiTrap column was washed with five more CV of Lysis Buffer before transferring to an ÄKTA fast protein liquid chromatography (FPLC) system (GE Healthcare) using solvent A (LahT150 Lysis Buffer) as stationary phase and solvent B (LahT150 Elution Buffer, 20 mM Tris, 1 M NaCl, 500 mM imidazole, pH 7.8 at 25°C) as mobile phase. The gradient increased linearly from 0% Solvent B to 20% Solvent B in Solvent A at a flow rate of 2.0 mL/min over 10 CV, followed by a linear increase from 20% to 100% Solvent B over 4 CV during which the protein eluted, and a final wash step at 100% Solvent B for 8 CV. The fractions with 280 nm absorbance were analyzed by SDS-PAGE, fractions containing the desired protein were concentrated to ~3 mL, and then applied to a HiLoad 16/60 gel filtration column containing Superdex200 resin (GE Healthcare). The column was eluted with 120 mL (2 CV) LahT150 storage buffer (10% glycerol, 20 mM Tris, 1 M NaCl, pH 7.8) and the fractions containing protein were concentrated to 4–6 mg/mL and aliquoted. Final yields varied between 40 and 50 mg/L of culture.

### In vitro protease assays

LahT150 cleavage assays contained 100 µM substrate and 10 µM LahT150 in 50 mM Tris pH 8.0. The cleavage assays produced identical results with and without the presence of 5 mM (tris(2-carboxyethyl)phosphine) (TCEP) and with as low as 1 µM LahT150. For mass spectrometry details, see *Table 1*.

### Bioinformatics

The top 5000 homologous sequences were retrieved using a BLASTp search against non-redundant protein sequences database using full length LahT sequence as query. The LahT sequences were imported into EFI-Enzyme Similarity Tool webserver using option C with FASTA header reading to construct the SSN (*Gerlt et al., 2015*). The 90% representative node network was opened and analyzed by Cytoscape, and 45% sequence identity was applied to separate the clusters. Nodes are shown in different colors based on the phylum classification, and several characterized LahT homologs are marked with red circles and their names shown on the side.

### Crystallization and structure determination

Repeated attempts to produce crystals of LahT150 failed, presumably due to the presence of the His$_6$ tag, as well as other potentially flexible regions. Multiple sequence alignments and secondary structure prediction analysis suggest that the last three C-terminal residues in the LahT150 construct were flexible. A new construct, encompassing residues 1–147 of LahT, was generated using the PCR and cloned into pET His6 TEV LIC cloning vector using HiFi DNA Assembly Master Mix for protein overexpression. The resultant plasmid (verified by sequencing) was used to transform chemically competent *E. coli* BL21 Rosetta 2 cells for overproduction. Starter cultures (each with 6 mL LB) were grown overnight and used to inoculate 2 L LB containing ampicillin (100 µg/mL) and chloramphenicol (25 µg/mL).

Cultures were grown at 37°C with vigorous shaking until the OD600 reached ~0.6 before cooling down in ice water for 15 min. Protein expression was induced by the addition of 0.5 mM IPTG and cultures were shaken for an additional 18 hr at 18°C and 200 rpm. Cell pellets were harvested by centrifugation at 4°C, resuspended with 40 mL suspension buffer (500 mM NaCl, 10% glycerol, 20 mM Tris, pH 8.0). Harvested cells were lysed by sonication, and the lysates were clarified by centrifugation at 4°C. The clear supernatant was loaded onto a 5 mL immobilized metal ion affinity resin column (Hi-Trap Ni-NTA, G.E. Healthcare) pre-equilibrated with binding buffer (1 M NaCl, 5% glycerol, 20 mM Tris, pH 8.0). The column was washed with 50 mL of 12% elution buffer (1 M NaCl, 250 mM imidazole, 20 mM Tris, pH 8.0), and then eluted by a linear gradient to 100% elution buffer. Fractions containing pure protein (as determined by SDS-PAGE) were combined and dialyzed against dialysis buffer (300 mM NaCl, 10% glycerol, 20 mM Tris-HCl, pH 7.5) overnight at 4°C. Purified proteins were concentrated, and the final concentration was quantified by Bradford analysis (Thermo Scientific), concentrated to ~10 mg/ml, and flash frozen in liquid nitrogen.

**Table 1.** Calculated and observed MALDI ToF $[M + H]^+$ masses for the leader peptides in *Figure 4—figure supplement 4*. n.d., not detected.

| $[M + H]^+$ | WT | V-4K | V-4T | V-4D | E-6A | E-6K | E-6D |
|---|---|---|---|---|---|---|---|
| Calcd | 8169.8 | 8227.0 | 8199.8 | 8229.8 | 8139.8 | 8168.9 | 8183.8 |
| Obsd | 8169.3 | 8229.9 | 8201.4 | n.d. | 8137.8 | 8170.0 | 8183.1 |
| | L-7A | L-7K | L-7D | E-8A | E-8K | E-8D | |
| Calcd | 8127.7 | 8212.9 | 8199.8 | 8111.8 | 8168.9 | 8155.8 | |
| Obsd | 8126.2 | n.d. | n.d. | 8111.0 | 8167.7 | 8157.5 | |
| | D-9A | D-10A | D-9E,D-10E | A-3Y | A-3F | A-3K | A-3E |
| Calcd | 8125.8 | 8125.8 | 8197.9 | 8261.9 | 8245.9 | 8226.9 | 8227.9 |
| Obsd | 8125.4 | 8127.0 | 8196.7 | 8261.1 | 8246.5 | 8225.8 | 8226.8 |
| | L-12A | L-12K | L-12D | L-12F | L-12W | L-12Y | |
| Calcd | 8127.7 | 8184.8 | 8171.7 | 8203.8 | 8242.8 | 8184.8 | |
| Obsd | 8126.0 | n.d. | n.d. | n.d. | 8241.0 | n.d | |

DOI: https://doi.org/10.7554/eLife.42305.013

Prior to crystallization, flash-frozen aliquots of recombinant, purified LahT150 were thawed and purified by size-exclusion chromatography (Superdex Hiload 75 16/60, GE Healthcare) using an iso-cratic gradient buffer composed of 100 mM KCl, 20 mM HEPES, pH 7.5 and concentrated. The puri-fied protein (8 mg/mL) was incubated with 1 mM of peptide aldehyde inhibitor for 2 hr before mixing with precipitant solution in a 1:1 ratio (v/v). The precipitant solution consisted of 0.02 M D-glucose, 0.02 M D-mannose, 0.02 M D-galactose, 0.02 M L-fucose, 0.02 M D-xylose, 0.02 M N-acetyl-D-glucosamine, 0.05 M Tris and BICINE pH 8.5, 20% v/v polyethylene glycol 500 mono-methyl ether, 10% w/v polyethylene glycol 20000, and 8% v/v formamide. Crystals of the protease-inhibitor appeared after 2 days incubation at 9℃, reached their largest size at 3–7 days, and were subsequently flash frozen by direct immersion into liquid nitrogen. All diffraction data were collected at LS-CAT (Sector 21, Advanced Photon Source, Argonne National Labs, IL) using MX-300 or Eiger 9M detectors. All data were integrated and scaled using either HKL2000 (*Minor et al., 2006*) or XDS (*Kabsch, 2014*).

Crystallographic phases were determined by the single wavelength anomalous diffraction method via AutoSol (*Terwilliger et al., 2009*) from data collected on a crystal soaked in precipitant solution with additional 2 mM 4-chloromercuribenzenesulfonate (PCMBS) for 2 hr. Phases for the native data set was determined using molecular replacement (*McCoy, 2007*). For each structure, iterative model building was carried out using the PHENIX suite of programs (*Afonine et al., 2012*) and further improved by manual rebuilding using COOT (*Emsley and Cowtan, 2004*). Cross-validation, using 5% of the data for the calculation of the free R factor (*Brunger, 2007*) was utilized throughout the model building process in order to monitor building bias. The stereochemistry of all of the mod-els was routinely monitored throughout the course of refinement using PROCHECK (*Laskowski et al., 1996*). Relevant data collection and refinement parameters are provided in *Table 2*. The coordinates for the LahT147-peptide structure can be accessed under PDB code 6MPZ.

## Cloning of LahT150

The N-terminal 450 nucleotides of the *lahT* gene harboring flanking sequences homologous to pET-Duet-1 multiple cloning site I (MCSI) were amplified from genomic DNA of *Lachnospiraceae* C6A11 as template and the primers LahT150_fp (5'- accatcatcaccacagccaggatccgaGTAAAAAGC AGA TACAGCCTGTCACAAGAG-3') and LahT150_rp (5'- tctgttcgacttaagcattatgcggccgcTTA CTG TTCAAATCTATCAGTAGGCTTG-3'). The pETDuet homology is displayed in lowercase letters. The PCR product was cloned into EcoRI/HindIII-linearized pETDuet by Gibson assembly (50℃, 1 hr) using a molar ratio of 10:1 (insert: backbone) (*Gibson et al., 2009*). The final construct was confirmed by DNA sequencing.

**Table 2.** Data collection, phasing and refinement statistics.

| | LahT-inhibitor 1 complex | PCMBS |
|---|---|---|
| Data collection | | |
| Space group | C2 | C2 |
| Unit cell (a,b,c,β) | 37.9, 119.4, 76.5, 93.8 | 37.3, 119.8, 83.5, 112.8 |
| Resolution | 76.4–1.98 (1.985–1.98) | 59.9–2.04 (2.05–2.04) |
| Total reflections | 239,058 | 124,854 |
| Unique reflections | 47,187 | 21,494 |
| $R_{sym}$ (%)[*] | 0.102 (0.727) | 0.090 (0.690) |
| $I/\sigma(I)$[*] | 9.3 (2.1) | 12.9 (2.5) |
| Completeness (%)[*] | 99.8 (99.8) | 99.9 (100) |
| Redundancy | 5.1 (5.1) | 5.9 (6.0) |
| Refinement | | |
| Resolution (Å) | 50.0–2.0 | |
| No. reflections | 43,389 | |
| $R_{work}$ / $R_{free}$[†] | 23.4/26.8 | |
| Number of atoms | | |
| Protein | 4479 | |
| Inh | 352 | |
| Water | 123 | |
| B-factors | | |
| Protein | 37.6 | |
| Inh | 34.5 | |
| Water | 35.9 | |
| R.m.s deviations | | |
| Bond lengths (Å) | 0.015 | |
| Bond angles (°) | 1.81 | |

[*]Highest resolution shell is shown in parenthesis.

[†]R-factor = $\Sigma(|F_{obs}|-k|F_{calc}|)/\Sigma |F_{obs}|$ and R-free is the R value for a test set of reflections consisting of a random 5% of the diffraction data not used in refinement.

DOI: https://doi.org/10.7554/eLife.42305.014

## Cloning of LahA substrates

The *lahA* genes were amplified using genomic DNA from *Lachnospiraceae* bacterium C6A11 as template and LahAx_fp and LahAx_rp as primers (*Table 3*) by touchdown PCR (*Korbie and Mattick, 2008*) with the annealing temperature decreasing from 70°C to 54°C over 80 cycles (−0.2°C/cycle). An example PCR amplification cycle consisted of denaturing (98°C for 10 s), annealing (from 70°C to 55°C, 0.2°C lower every cycle for a total of 80 cycles) for 30 s and extension (72°C for 30 s). Subsequently the PCR fragment was cloned by using Gibson assembly into the multiple cloning site 1 (MCSI) of the pRSFDuet-1 vector previously linearized by HindIII and EcoRI digestion.

## Cloning of AzoA substrates

AzoA2, 3, 6, and 7 were amplified from *Azospirillum* sp. B510 genomic DNA (JCM 14679) via PCR using the appropriate AzoA forward and reverse primers (*Table 4*) and ligated into the pRSFDuet-1 vector (Novagen) after digestion with the restriction enzymes BamHI and NotI (for AzoA2) or BamHI and HindIII (for AzoA3, 6, 7). The N- and C-terminally His₆-tagged *E. coli mbp* gene was amplified from the MBP-pET28 vector (a gift from Douglas A. Mitchell, University of Illinois at Urbana-Champaign) (*Lee et al., 2008*) using the appropriate MBP-AzoA G1/G4 primers. The gene encoding MBP was introduced such that MBP was appended to the N-termini of the AzoAs via Gibson assembly

**Table 3.** Primers used in the generation of LahA constructs.
Homology with vector backbone is displayed as lowercase letters.

| Primer Name | Sequence 5′−3′ |
|---|---|
| LahA1_fp | accatcatcaccacagccaggatccgaattcgaACGAGAATTTAGAGAAGTTTTTTCAGA |
| LahA1_rp | ttctgttcgacttaagcattatgcggccgcAGATTGCTCCTGCAGCGAAATTGGTAAG |
| LahA2_fp | accatcatcaccacagccaggatccgaattcgaACGAGAATTTAAAGATGTTTTTGCAGA |
| LahA2_rp | ttctgttcgacttaagcattatgcggccgcTTAGATTGCTGTTGCAGCGAAAAGGGAAT |
| LahA3_fp | accatcatcaccacagccaggatccgaattcgaATGATAGTTTAAAAGAGTTTTTGAA |
| LahA3_rp | ttctgttcgacttaagcattatgcggccgcTTAGACGGCTCCGGCTGACGATGCCGCAA |
| LahA4_fp | accatcatcaccacagccaggatccgaattcgaACGAGAATTTAAAGATGTTTTTACAGA |
| LahA4_rp | ttctgttcgacttaagcattatgcggccgcTTAAACCGCAAGTAAACTCATCGTTACAGC |
| LahA5_fp | accatcatcaccacagccaggatccgaattcgaACGAGAATCTCAAGCTATTTTTACAA |
| LahA5_rp | ttctgttcgacttaagcattatgcggccgcTTACATTGCCGATAATGATAATGATAATGC |
| LahA6_fp | accatcatcaccacagccaggatccgaattcgaATGAAAGGATAAAAGATTTATTTACCG |
| LahA6_rp | ttctgttcgacttaagcattatgcggccgcTTACATAAGTGCCTTTCTTATTGCAGTAAG |
| LahA7_rp | accatcatcaccacagccaggatccgaattcgaACGAGAACTTGAAGAAATTCCTGGAGG |
| LahA7_fp | ttctgttcgacttaagcattatgcggccgcTTATGAAGCAATCCTTGACCAACTATTGA |

DOI: https://doi.org/10.7554/eLife.42305.015

with the linearized AzoA-pRSFDuet vectors (amplified using MBP-AzoA G2/G3 primers). A sequence encoding a C-terminal tag with the amino acid sequence DAHHHHHH was added to the AzoA2, 3, and 7 constructs via overlap-extension PCR with the MBP-AzoACHis G1/G2 primers and one-component Gibson assembly (*Gibson et al., 2009*).

## Cloning of XY33a substrates

The gene encoding XY33a was amplified from a pRSFDuet plasmid containing XY33a in MCSI and ProcM in MCSII (*Yang et al., 2018*) using primers XY33A_fp and XY33A_rp and touchdown PCR (*Korbie and Mattick, 2008*) with the annealing temperature decreasing from 70°C to 54°C over 80 cycles (−0.2°C/cycle). An example PCR amplification cycle consisted of denaturing (98°C for 10 s), annealing (from 70°C to 55°C, 0.2°C lower every cycle for a total of 80 cycles) for 30 s and extension (72°C for 30 s). The genes encoding XY33a variants A−3Y, A−3F, A−3K and A−3E, V−4K, V−4T, and V−4D, L−12A, L−12K, L−12D, L−12F, L−12W, L-12Y were amplified using the same protocol with the purchased synthetic genes in *Table 5* as templates. All PCR products above, containing homology for Gibson assembly (*Gibson et al., 2009*) into an EcoRI/HindIII pRSFDuet digest, were purified by agarose gel electrophoresis and extracted using a Qiaquick Gel Extraction kit. The inserts were assembled into the EcoRI/HindIII-linearized pRSFDuet backbone using a molar ratio of 10:1 (insert: backbone) using the Gibson method (*Gibson et al., 2009*). The final construct was confirmed by sequencing. XY33a mutants D−10E/D−9E, D−9A, D−10A, E−8A, E−8K, E−8D, L−7A, L−7K, L−7D, E−6A, E−6K, and E−6D were generated in the following manner. The XY33a-wild type (MCSI) pRSFDuet plasmid was used as a template for two PCR reactions. One reaction used the RSF_fp primer and the gene-specific reverse primer encoding the mutation, and the other reaction used the RSF_rp primer and the forward primer encoding the desired mutation. The PCR amplification cycles were identical to the ones used for amplifying the gene encoding wild-type XY33a, except the extension time was 150 s. This strategy splits the pRSFDuet-1 vector in two roughly equal parts for Gibson assembly. The two PCR products were gel purified and extracted using a Qiaquick Gel Extraction kit and used in a Gibson assembly reaction in equimolar amounts. The Gibson reactions were used to transform *E. coli* DH10b, and the identities of the resulting constructs was verified by sequencing.

**Table 4.** Primers used in the cloning of AzoA constructs.
Homology with vector backbone is displayed as lowercase letters.

| Primer name | Sequence 5′−3′ |
| --- | --- |
| AzoA2 fwd | aaaGGATCCatgacaaccgaaacgcaaacc |
| AzoA2 rev | aaaGCGGCCGCctaccattttctgggaatggccaag |
| AzoA3 fwd | caatggacggtGGATCCGatgacagaccaaacccagtccacatcc |
| AzoA3 rev | cggaaacagccAAGCttactgttgtcgcaaacgcggtggtga |
| AzoA6 fwd | aaaggacttcgGGATCCgatgacaaatgaaacgcagcccacc |
| AzoA6 rev | ttatgggatcCAAGCTTctaccatttcctcgttccgagaatggc |
| AzoA7 fwd | caatggacccaGGATCCgatgacagaccaaacgcagtccgcc |
| AzoA7 rev | catggacatcCAAGCTTctaccattttgcacacacccccctgat |
| MBP-AzoA G1 | aataaggagatataccatgGGCAGCAGCCATCATCATCATC |
| MBP-AzoA G2 | TGGCTGCTGCCcatggtatatctccttattaaagttaaacaaaattatttctacagggg |
| MBP-AzoA2 G3 | CTGTACTTCCAATCCatgacaaccgaaacgcaaaccgcc |
| MBP-AzoA2 G4 | cgtttcggttgtcatGGATTGGAAGTACAGGTTCTCAGATCCACGC |
| MBP-AzoA3 G3 | CTGTACTTCCAATCCatgacagaccaaacccagtccac |
| MBP-AzoA3 G4 | ggtttggtctgtcatGGATTGGAAGTACAGGTTCTCAGATCCACGC |
| MBP-AzoA6 G3 | CTGTACTTCCAATCCatgacaaatgaaacgcagccc |
| MBP-AzoA6 G4 | cgtttcatttgtcatGGATTGGAAGTACAGGTTCTCAGATCCACGC |
| MBP-AzoA7 G3 | CTGTACTTCCAATCCatgacagaccaaacgcagtccgcc |
| MBP-AzoA7 G4 | gcgtttggtctgtcatGGATTGGAAGTACAGGTTCTCAGATCCACGC |
| AzoA2CHis G1 | tctaGTGATGGTGATGGTGATGTGCATCccattttctgggaatggccaagc |
| AzoA2CHis G2 | GATGCACATCACCATCACCATCACtagaagcttgcggccgcataatgcttaagtcg |
| AzoA3CHis G1 | tctaGTGATGGTGATGGTGATGTGCATCctgttgtcgcaaacgcggtggtg |
| AzoA3CHis G2 | GATGCACATCACCATCACCATCACtagaagcttgcggccgcataatgcttaagtcg |
| AzoA7CHis G1 | tctaGTGATGGTGATGGTGATGTGCATCccattttgcacacacccccctgattccacc |
| AzoA7CHis G2 | GATGCACATCACCATCACCATCACtagaagcttgcggccgcataatgcttaagtcg |

DOI: https://doi.org/10.7554/eLife.42305.016

## Expression and purification of MBP-tagged AzoA2, 3, 6 and 7

MBP-AzoA-pRSFDuet constructs were used to transform *E. coli* Rosetta 2 (Novagen). Overnight cultures were diluted 1:100 into 1–2 L of LB containing kanamycin (50 µg/mL) and chloramphenicol (34 µg/mL), grown aerobically at 37°C (200 rpm) to $OD_{600}$ 0.6 and induced with 250 µM isopropyl β-D-1-thiogalactopyranoside (IPTG). Induction was allowed to proceed at 18°C (200 rpm) for 1–5 days. Cells were harvested at 8,000 × g and resuspended in 30–60 mL of lysis buffer (25 mM Tris pH 8.0, 500 mM NaCl, 20 mM imidazole, 10% [v/v] glycerol) containing 0.2–0.3 mg/mL lysozyme (Gold Biotech) and ¼ of a protease inhibitor tablet (Roche cOmplete, EDTA-free) for 1–2 hr before lysis by sonication for 8–9 × 60 s while stirring on ice. Lysate was clarified by centrifugation at 38,000 × g and loaded onto columns containing 2–5 mL of HisPur Ni-NTA Superflow agarose (Thermo Scientific) equilibrated in lysis buffer. The resin was washed with 30–50 mL each of lysis buffer and wash buffer (lysis buffer with 40 mM imidazole) before elution with 10–15 mL elution buffer (lysis buffer with 300 mM NaCl and 200 mM imidazole). Eluted protein was concentrated to <2.5 mL using a centrifugal concentrator (EMD Millipore) and exchanged into storage buffer (25 mM Tris pH 8.0, 500 mM NaCl, 10% [v/v] glycerol) using a PD-10 desalting column (GE Healthcare), concentrated to <500 µL, flash frozen, aliquoted, and stored at −80°C until use. Protein concentration was estimated by absorbance at 280 nm using extinction coefficients calculated by ExPASy (http://web.expasy.org/protparam); yields ranged from 5 to 15 mg/L culture.

**Table 5.** Primers and synthetic genes used in the cloning of XY33a constructs.
Mutations are shown in bold font. Homology with vector backbone is displayed as lowercase letters.

| Primer or synthetic gene name | Sequence (5′—3′) |
| --- | --- |
| XY33A_fp | catcaccatcatcaccacagccaggatccGTCTGAAGAGCAACTGAAGGC |
| XY33A_rp | gtacaatacgattactttctgttcgacttaagcattatTTAGCAAATATCGAGGACGTG |
| RSF_fp | Gcaggcgttttttccatagg |
| RSF_rp | Ctggcttgagcgtcgattttttg |
| XY33a_D-9A_fp | CGCCAAAATCTGTCTGAA **GCA** AGCTGGAAGGTGTGGC |
| XY33a_D-9A_rp | GCCACACCTTCCAGCT **TGC** TTCAGACAGATTTTGGCG |
| XY33a_D-10A_fp | CGCCAAAATCTGTCT **GCA** GAAAGCTGGAAGGTGTGGC |
| XY33a_D-10A_rp | GCCACACCTTCCAGCTTTC **TGC** AGACAGATTTTGGCG |
| XY33a_D-10E,D-9E_fp | CGCCAAAATCTGTCT GAA **GAA** AGCTGGAAGGTGTGGCTG |
| XY33a_D-10E,D-9E_rp | CAGCCACACCTTCCAGCT TTC **TTC** AGACAGATTTTGGCG |
| XY33a_E-8A_fp | CAAAATCTGTCTGATGAT **GCA** CTGGAAGGTGTGGCTGGG |
| XY33a_E-8A_rp | CCCAGCCACACCTTCCAG **TGC** ATCATCAGACAGATTTTG |
| XY33a_E-8K_fp | CAAAATCTGTCTGATGAT **AAA** CTGGAAGGTGTGGCTGGG |
| XY33a_E-8K_rp | CCCAGCCACACCTTCCAG **TTT** ATCATCAGACAGATTTTG |
| XY33a_E-8D_fp | CAAAATCTGTCTGATGAT **GAT** CTGGAAGGTGTGGCTGGG |
| XY33a_E-8D_rp | CCCAGCCACACCTTCCAG **ATC** ATCATCAGACAGATTTTG |
| XY33a_L-7A_fp | CTGTCTGATGATGAG **GCA** GAAGGTGTGGCTGGGG |
| XY33a_L-7A_rp | CCCCAGCCACACCTTC **TGC** CTCATCATCAGACAG |
| XY33a_L-7K_fp | CTGTCTGATGATGAG **AAA** GAAGGTGTGGCTGGGG |
| XY33a_L-7K_rp | CCCCAGCCACACCTTC **TTT** CTCATCATCAGACAG |
| XY33a_L-7D_fp | CTGTCTGATGATGAG **GAT** GAAGGTGTGGCTGGGG |
| XY33a_L-7D_rp | CCCCAGCCACACCTTC **ATC** CTCATCATCAGACAG |
| XY33a_E-6A_fp | GTCTGATGATGAGCTG **GCA** GGTGTGGCTGGGGGAG |
| XY33a_E-6A_rp | CTCCCCCAGCCACACC **TGC** CAGCTCATCATCAGAC |
| XY33a_E-6K_fp | GTCTGATGATGAGCTG **AAA** GGTGTGGCTGGGGGAG |
| XY33a_E-6K_rp | CTCCCCCAGCCACACC **TTT** CAGCTCATCATCAGAC |
| XY33a_E-6D_fp | GTCTGATGATGAGCTG **GAT** GGTGTGGCTGGGGGAG |
| XY33a_E-6D_rp | CTCCCCCAGCCACACC **ATC** CAGCTCATCATCAGAC |
| XY33a_V-4K_gene | TCTGAAGAGCAACTGAAGGCATTCCTCACCAAAGTTCAA GCCGATACTTCACTACAGGAACAGTTAAAGATAGAAGGA GCTGATGTTGTAGCCATTGCCAAAGCTGTAGGCTTCTCG ATTACCACAGAAGACCTAAACTCTCATCGCCAAAATCTG TCTGATGATGAGCTGGAAGGT**AAA**GCTGGGGGAGCGG CCTGTCATTTCCTTCTTTTCTCTATGCCTCCATCCCA CGTCCTCGATATTTGCTAA |
| XY33a_V-4T_gene | TCTGAAGAGCAACTGAAGGCATTCCTCACCAAAGTT CAAGCCGATACTTCACTACAGGAACAGTTAAAGATA GAAGGAGCTGATGTTGTAGCCATTGCCAAAGCTG TAGGCTTCTCGATTACCACAGAAGACCTAAACTC TCATCGCCAAAATCTGTCTGATGATGAGCTGGAA GGT**ACC**GCTGGGGGAGCGGCCTGTCATTTCCTT CTTTTCTCTATGCCTCCATCCCACGTCCTC GATATTTGCTAA |
| XY33a_V-4D_gene | TCTGAAGAGCAACTGAAGGCATTCCTCACC AAAGTTCAAGCCGATACTTCACTACAGGAA CAGTTAAAGATAGAAGGAGCTGATGTTGT AGCCATTGCCAAAGCTGTAGGCTTCTCGAT TACCACAGAAGACCTAAACTCTCATCGCCAA AATCTGTCTGATGATGAGCTGGAAGGT**GAT**GCTGGGGGAGCGGCCTGTCATTTCCTTCT TTTCTCTATGCCTCCATCCCACGTCCT CGATATTTGCTAA |

*Table 5 continued*

| Primer or synthetic gene name | Sequence (5′—3′) |
|---|---|
| XY33a_A-3Y_gene | TCTGAAGAGCAACTGAAGGCATTCCTCA CCAAAGTTCAAGCCGATACTTCACTACAG GAACAGTTAAAGATAGAAGGAGCTGATGT TGTAGCCATTGCCAAAGCTGCAGGCTTCT CGATTACCACAGAAGACCTAAACTCTCAT CGCCAAAATCTGTCTGATGATGAGCTGGA AGGTGTG**TAT**GGGGGAGCGGCCTGTCATT TCCTTCTTTTCTCTATGCCTCCATCCCACG TCCTCGATATTTGCTAA |
| XY33a_A-3F_gene | TCTGAAGAGCAACTGAAGGCATTCCTCAC CAAAGTTCAAGCCGATACTTCACTACAGG AACAGTTAAAGATAGAAGGAGCTGATGTT GTAGCCATTGCCAAAGCTGCAGGCTTCT CGATTACCACAGAAGACCTAAACTCTCAT CGCCAAAATCTGTCTGATGATGAGCTGGA AGGTGTG**TTT**GGGGGAGCGGCCTGTCAT TTCCTTCTTTTCTCTATGCCTCCATCCCA CGTCCTCGATATTTGCTAA |
| XY33a_A-3E_gene | TCTGAAGAGCAACTGAAGGCATTCCTCA CCAAAGTTCAAGCCGATACTTCACTACAG GAACAGTTAAAGATAGAAGGAGCTGATG TTGTAGCCATTGCCAAAGCTGCAGGCT TCTCGATTACCACAGAAGACCTAAACTC TCATCGCCAAAATCTGTCTGATGATGAGC TGGAAGGTGTG**GAA**GGGGGAGCGGCCT GTCATTTCCTTCTTTTCTCTATGCCTCC ATCCCACGTCCTCGATATTTGCTAA |
| XY33a_A-3K_gene | TCTGAAGAGCAACTGAAGGCATTCCTCA CCAAAGTTCAAGCCGATACTTCACTACAG GAACAGTTAAAGATAGAAGGAGCTGATG TTGTAGCCATTGCCAAAGCTGCAGGCTT CTCGATTACCACAGAAGACCTAAACTCT CATCGCCAAAATCTGTCTGATGATGAGC TGGAAGGTGTG**AAA**GGGGGAGCGGC CTGTCATTTCCTTCTTTTCTCTATGCC TCCATCCCACGTCCTCGATATTTGCTAA |
| XY33a_L-12A_gene | TCTGAAGAGCAACTGAAGGCATTCCTC ACCAAAGTTCAAGCCGATACTTCACTA CAGGAACAGTTAAAGATAGAAGGAGC TGATGTTGTAGCCATTGCCAAAGCTG CAGGCTTCTCGATTACCACAGAAGAC CTAAACTCTCATCGCCAAAAT**GCG**TC TGATGATGAGCTGGAAGGTGTGGCT GGGGGAGCGGCCTGTCATTTCCTTC TTTTCTCTATGCCTCCATCCCACGTCC TCGATATTTGCTAA |
| XY33a_L-12K_gene | TCTGAAGAGCAACTGAAGGCATTCCTCA CCAAAGTTCAAGCCGATACTTCACTACAGG AACAGTTAAAGATAGAAGGAGCTGATGTTG TAGCCATTGCCAAAGCTGCAGGCTTCTCGA TTACCACAGAAGACCTAAACTCTCATCGCC AAAAT**AAA**TCTGATGATGAGCTGGAAGGT GTGGCTGGGGGAGCGGCCTGTCATTTCC TTCTTTTCTCTATGCCTCCATCCCACGTC CTCGATATTTGCTAA |
| XY33a_L-12D_gene | TCTGAAGAGCAACTGAAGGCATTCCTCA CCAAAGTTCAAGCCGATACTTCACTACA GGAACAGTTAAAGATAGAAGGAGCTGA TGTTGTAGCCATTGCCAAAGCTGCAGG CTTCTCGATTACCACAGAAGACCTAAAC TCTCATCGCCAAAAT**GAT**TCTGATGATG AGCTGGAAGGTGTGGCTGGGGGAGCG GCCTGTCATTTCCTTCTTTTCTCTATGCC TCCATCCCACGTCCTCGATATTTGCTAA |

*Table 5 continued on next page*

*Table 5 continued*

| Primer or synthetic gene name | Sequence (5′−3′) |
| --- | --- |
| XY33a_L-12F_gene | TCTGAAGAGCAACTGAAGGCATTCCTCA<br>CCAAAGTTCAAGCCGATACTTCACTACAG<br>GAACAGTTAAAGATAGAAGGAGCTGATGT<br>TGTAGCCATTGCCAAAGCTGCAGGCTTCTC<br>GATTACCACAGAAGACCTAAACTCTCATCG<br>CCAAAAT**TTTT**CTGATGATGAGCTGGAAGG<br>TGTGGCTGGGGGAGCGGCCTGTCATTTC<br>CTTCTTTTCTCTATGCCTCCATCCCACG<br>TCCTCGATATTTGCTAA |
| XY33a_L-12W_gene | TCTGAAGAGCAACTGAAGGCATTCCTCA<br>CCAAAGTTCAAGCCGATACTTCACTACA<br>GGAACAGTTAAAGATAGAAGGGAGCTGATG<br>TTGTAGCCATTGCCAAAGCTGCAGGCTTC<br>TCGATTACCACAGAAGACCTAAACTCTCA<br>TCGCCAAAAT**TGG**TCTGATGATGAGCTG<br>GAAGGTGTGGCTGGGGGAGCGGCCTGTC<br>ATTTCCTTCTTTTCTCTATGCCTCCATCC<br>CACGTCCTCGATATTTGCTAA |
| XY33a_L-12Y_gene | TCTGAAGAGCAACTGAAGGCATTCCTCA<br>CCAAAGTTCAAGCCGATACTTCACTACA<br>GGAACAGTTAAAGATAGAAGGGAGCTGATG<br>TTGTAGCCATTGCCAAAGCTGCAGGCTT<br>CTCGATTACCACAGAAGACCTAAACTCTC<br>ATCGCCAAAAT**TAT**TCTGATGATGAGCTG<br>GAAGGTGTGGCTGGGGGAGCGGCCTGT<br>CATTTCCTTCTTTTCTCTATGCCTCCAT<br>CCCACGTCCTCGATATTTGCTAA |

DOI: https://doi.org/10.7554/eLife.42305.017

## Expression and purification of XY33a, ProcA, and LahA

*E. coli* BL21 (DE3) cells were transformed with pRSFDuet-1 plasmids encoding either the N-terminally His-tagged ProcA 2.8 variants or LahAs in MCSI. An overnight culture was added to a culture flask containing TB with 2% glucose (1:50; v/v; overnight culture: overexpression culture), kanamycin (50 µg/mL) and 2.0 mM MgCl$_2$. The culture was incubated in a 37°C shaker until OD$_{600}$ reached 1.2–1.5. The cultures were cooled to 22°C and IPTG (500 µM final concentration for ProcA and XY33a peptides, 250 µM for LahA peptides) was used to induce expression. Following 16–20 hr incubation at 22°C, the cells were harvested at 5000 × g for 10 min and resuspended in LanA B1 Buffer (6.0 M guanidine hydrochloride, 0.5 mM imidazole, 20 mM NaH$_2$PO$_4$, pH 7.5), using 30–50 mL of LanA B1 Buffer for each liter of culture. Resuspended cells were stored at −80°C until purification. Freeze-thawing in 6.0 M guanidine hydrochloride lead to lysis of the cells, and the thawed cells were directly centrifuged at 30,000 × g for 30 min at 4°C. The supernatants were applied to 2–3 mL of His60 Clontech Ni superflow resin (catalog number 635660) that had been charged with two column volumes (CV) of 0.1 M NiSO$_4$, washed with 10 CV of water and equilibrated with 10 CV of LanA B1 Buffer. The column was washed with ten CV of LanA B2 Buffer (4.0 M guanidine hydrochloride, 20 mM NaH$_2$PO$_4$, 30 mM imidazole, 300 mM NaCl, pH 7.5). Then, between 5–7 CV of LanA Elute Buffer (4.0 M guanidine hydrochloride, 20 mM TrisHCl, 1.0 M imidazole, 100 mM NaCl, pH 7.5) was used to elute the peptide. Peptides were desalted by SPE using an Agilent Bond Elut C18 SPE column following the manufacturer instructions and lyophilized. Dry peptides were resuspended in 5–10% Solvent B (0.1% TFA in MeCN) and purified by RP-HPLC using a Phenomenex C5 column (5 µm, 100 Å, 250 mm ×10 mm) with a linear gradient from 2% Solvent B in Solvent A (0.1% TFA in water) to 100% Solvent B in 45 min, monitoring absorbance at 220 nm. Fractions containing His-tagged peptide were identified by MALDI-TOF MS and lyophilized. Final yields varied between 3–5 mg/mL His-tagged peptide per liter of culture.

## Synthesis of inhibitor 1

ProTide Cl-TCP Cl resin (CEM) was used for the solid phase peptide synthesis (SPPS) of the ProcA2.8 sequence-based aldehyde inhibitor on a 0.2 mmol scale. The resin was suspended in 5 mL

dichloromethane (DCM) at 0°C and $SOCl_2$ (1.2 equiv.) and pyridine (2.4 equiv.) were added. The resin was stirred at reflux for 3 hr, filtered through a fritted funnel, washed with DCM and dried for 20 min under vacuum. FmocGly (five equiv.) was dissolved in 3 mL DCM in a round bottom flask and diisopropylethylamine (DIPEA, 8 equiv.) was added and the resulting solution was stirred for 20 min at room temperature. The dried resin was added to the round bottom flask and the reaction was stirred overnight at room temperature. The resin was then transferred to a fritted funnel, washed with DCM and dried under vacuum, then capped twice for 10 min using DCM:MeOH:DIPEA (80:15:5 v/v/v) as capping solution while sparging with nitrogen. After capping, the resin was washed with DCM and dried under vacuum. Fmoc-Gly-loaded-resin (7 mg) were deprotected using 3 mL of 20% piperidine in DMF. Loading was determined by absorbance at 290 nm measured on a NanoDrop 2000 (ThermoFisher), using 20% piperidine in DMF as a blank. SPPS conditions involved 0.2 M Fmoc-protected amino acid in DMF, 0.5 M (7-azabenzotriazol-1-yloxy)tripyrrolidinophosphonium hexafluorophosphate (PyAOP) and 0.5 M 1-hydroxybenzotriazole (HOBt) as activator, 2 M DIPEA as activator base, and 20% piperidine in DMF with 0.1 M HOBt as deprotection solution. After coupling of the first amino acid (Ala−2), the remainder of the synthesis (residues −3 through −13) was performed on a CEM Liberty Microwave peptide synthesizer. Final Fmoc deprotection was performed under microwave activation, then the resin was transferred to a glass fritted funnel and washed with DCM and dried under vacuum for 20 min. The resin was then transferred to a round bottom flask and N-terminally acetylated using a solution of acetic anhydride:pyridine (5 mL, 1:2 v/v) for 1 hr at room temperature. The resin was again transferred to a fritted funnel, washed with DCM, dried under vacuum for 20 min, and then transferred to a clean round bottom flask. Selective cleavage of the fully protected peptide was performed with 20% hexafluoroisopropanol in DCM for 1 hr at room temperature. This step cleaves the fully protected peptide from the resin leaving only the C-terminal carboxylic acid available for further reaction. This solution was filtered into a clean oven-dried round bottom flask and concentrated under vacuum. The peptide was dissolved in 5 mL anhydrous DCM under a nitrogen atmosphere. Aminoacetaldehyde (three equiv.), PyAOP (three equiv.) and N-methylmorpholine (10 equiv.) were mixed in 3 mL anhydrous DCM under nitrogen, then slowly added to the fully protected peptide solution over 30–40 min. The reaction was stirred at room temperature overnight, then concentrated by rotovap before total deprotection using a mixture of trifluoroacetic acid:water (95:5 v/v) for 1 hr at room temperature. The use of triisopropylsilane (TIPS) resulted in the reduction of the C-terminal aldehyde to the corresponding alcohol and hence TIPS was not added to the deprotection cocktail. The peptide was precipitated by adding 10-fold excess (v/v) of cold diethylether, centrifuged at $10,000 \times g$ for 20 min, the supernatant removed, and the peptide dried under a nitrogen stream. The overall yield of the Ac-NLSDDELEGVAGG(aldehyde) peptide was 8% after RP-HPLC of the crude material resulting from SPPS, using a linear gradient from 1% Solvent B (0.1% TFA in acetonitrile) in Solvent A (0.1% TFA in water) to 61% solvent B over 61 min.

## Acknowledgements

This work was supported by grants from the National Institutes of Health (GM R37 058822 to WAV and GM079038 to SKN), and MINECO/FEDER (CTQ2015-70524-R and RYC-2013–14706 grants to GJO). NM acknowledges Universidad de La Rioja for a predoctoral fellowship. We thank Dr. William Kelly (AgResearch, New Zealand) for providing *Lachnospiraceae bacterium* C6A11, and Keith Brister and colleagues for facilitating data collection at LS-CAT (Argonne National Labs, IL).

## Additional information

### Competing interests

Wilfred A van der Donk: Reviewing editor, *eLife*. The other authors declare that no competing interests exist.

### Funding

| Funder | Grant reference number | Author |
|---|---|---|
| National Institutes of Health | GM058822 | Wilfred A van der Donk |

| Ministerio de Economía y Competitividad | CTQ2015-70524-R | Gonzalo Jiménez-Osés |
| National Institutes of Health | GM079038 | Satish K Nair |
| Ministerio de Economía y Competitividad | RYC-2013-14706 | Gonzalo Jiménez-Osés |
| Universidad de La Rioja | Predoctoral fellowship | Nuria Mazo |

The funders had no role in study design, data collection and interpretation, or the decision to submit the work for publication.

## Author contributions
Silvia C Bobeica, Formal analysis, Investigation, Visualization, Methodology, Writing—review and editing; Shi-Hui Dong, Martin I McLaughlin, Formal analysis, Investigation, Visualization, Writing—review and editing; Liujie Huo, Conceptualization, Formal analysis, Investigation, Visualization, Methodology; Nuria Mazo, Formal analysis, Investigation; Gonzalo Jiménez-Osés, Conceptualization, Formal analysis, Supervision, Funding acquisition, Investigation, Writing—review and editing; Satish K Nair, Conceptualization, Resources, Formal analysis, Supervision, Funding acquisition, Investigation, Visualization, Writing—original draft, Writing—review and editing; Wilfred A van der Donk, Conceptualization, Formal analysis, Supervision, Funding acquisition, Investigation, Visualization, Writing—original draft, Writing—review and editing

## Author ORCIDs
Silvia C Bobeica (iD) http://orcid.org/0000-0001-5058-5543
Shi-Hui Dong (iD) http://orcid.org/0000-0002-1743-2163
Martin I McLaughlin (iD) http://orcid.org/0000-0003-4410-0786
Gonzalo Jiménez-Osés (iD) http://orcid.org/0000-0003-0105-4337
Satish K Nair (iD) http://orcid.org/0000-0003-1790-1334
Wilfred A van der Donk (iD) http://orcid.org/0000-0002-5467-7071

## Decision letter and Author response
Decision letter https://doi.org/10.7554/eLife.42305.022
Author response https://doi.org/10.7554/eLife.42305.023

# Additional files

## Supplementary files
• Transparent reporting form
DOI: https://doi.org/10.7554/eLife.42305.018

## Data availability
Diffraction data has been deposited at Protein Data Bank under 6MPZ.

The following dataset was generated:

| Author(s) | Year | Dataset title | Dataset URL | Database and Identifier |
| --- | --- | --- | --- | --- |
| Dong SH, Nair SK | 2019 | Crystal structure of a double glycine motif protease from AMS/PCAT transporter in complex with the leader peptide | http://www.rcsb.org/structure/6MPZ | Protein Data Bank, 6MPZ |

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
