## [Decision Letter]

Thank you for submitting your article "Insights into AMS/PCAT Transporters from Biochemical and Structural Characterization of a Double Glycine Motif Protease" for consideration by *eLife*. Your article has been reviewed by two peer reviewers, and the evaluation has been overseen by a Reviewing Editor and John Kuriyan as the Senior Editor. The following individuals involved in review of your submission have agreed to reveal their identity: Jue Chen (Reviewer #1); Charles S. Craik (Reviewer #2).

The reviewers have discussed the reviews with one another and the Reviewing Editor has drafted this decision to help you prepare a revised submission.

Both reviewers and the Reviewing Editor agreed that your manuscript describes an important and interesting advance in our understanding the interplay between proteolysis and transport of peptide-based natural products based on the structural and biochemical determinants. Some concerns were raised about the presentation of results in the figures that should be addressed to ensure that the scientific findings can be fully appreciated with clarify by the general reader.

Summary:

The authors have combined their expertise in microbiology and structural biology to address an important question in biology: how PCAT transporters recognize their peptide substrates. First, the authors identified and characterized a PCAT transporter that transports a group of peptides (most of PCATs only transport one or two substrates). Then they determined the crystal structure of the protease domain in complex with a substrate, which provides atomic description of substrate recognition. The experiments were carried out at high standards and the paper is generally well written.

Essential revisions:

1) Some of the figures do not do the work justice and half of them made the process confusing for figuring out what and why something was done. It seemed that two different people worked independently on the figures and the text. This is particularly the case for Figure 4D where I might have missed that in the main text, but I could not find the part where it's referenced even though I think I figured out what they're trying to show there. I feel that Figure 3—figure supplement 5 is essential to judge the strength of their claim regarding broad specificity towards the core peptide. I suggest moving that into the main figure and removing some of the less important panels elsewhere.

2) Figure 1A. I am confused about what this figure is trying to achieve. The description only states it illustrates "posttranslational modifications" and the main text refers to Figure 1A for "posttranslational modification enzymes often act iteratively on a subset of amino acids in a C-terminal core peptide in a process that is directed by an N-terminal leader peptide". I feel that the figure needs to be reworked, elucidating what kind of modifications are supposed to be happening in that panel and what the meaning of the color scheme is or what role the leader peptide plays using the crystal structure in the same manuscript. Alternatively, it can be removed since I do not think the reader will have a conceptual problem regarding the meaning of "removal of the leader peptide" which is well known and not in need of illustration.

3) Figure 1B. While I appreciate the high sequence similarity between the leader peptide candidates, I have a more difficult time appreciating the sequence diversity on the core peptide. An additional panel highlighting the diversity of amino acids accepted after the double Gly motif would be helpful.

4) Figure 1C. I am confused by Figure 1C since I could not find a reference in the main text until the last paragraph of the subsection “Identification of a substrate tolerant protease”. The first reference in "Results" refers to Figure 2. It is not self-explanatory why mass spec data showing proteolytic activity is shown here since leader peptide removal and inhibitor design receive a dedicated figure with Figure 3.

5) Figure 4A and D. In both cases it is unclear to me why these panels exist. The main text takes no note of them which is particularly unfortunate in the case of Figure 4D. The main text notes an apparent narrowing of the substrate binding pocket towards the double Gly motif and the figure suggests this to be accompanied by an increase in hydrophobicity.

---

## [Author Response]

Essential revisions:1) Some of the figures do not do the work justice and half of them made the process confusing for figuring out what and why something was done. It seemed that two different people worked independently on the figures and the text. This is particularly the case for Figure 4D where I might have missed that in the main text, but I could not find the part where it's referenced even though I think I figured out what they're trying to show there. I feel that Figure 3—figure supplement 5 is essential to judge the strength of their claim regarding broad specificity towards the core peptide. I suggest moving that into the main figure and removing some of the less important panels elsewhere.

We agree that the presentation of the work can be improved, and indeed multiple authors and figure makers worked on this manuscript. We have tried to better integrate the figures into the text.

Regarding Figure 3, now Figure 4, we moved parts of the former figure supplements into the main figure as suggested.

2) Figure 1A. I am confused about what this figure is trying to achieve. The description only states it illustrates "posttranslational modifications" and the main text refers to Figure 1A for "posttranslational modification enzymes often act iteratively on a subset of amino acids in a C-terminal core peptide in a process that is directed by an N-terminal leader peptide". I feel that the figure needs to be reworked, elucidating what kind of modifications are supposed to be happening in that panel and what the meaning of the color scheme is or what role the leader peptide plays using the crystal structure in the same manuscript. Alternatively, it can be removed since I do not think the reader will have a conceptual problem regarding the meaning of "removal of the leader peptide" which is well known and not in need of illustration.

We agree that we were not very effective in delivering our intended message. In our view, our study has two main points, but they are very different. One main point is the molecular mechanism of substrate (i.e. leader peptide) recognition and that seems to have come through with the reviewers. The other message, which we do not seem to have been particularly successful with, is that this enzyme LahT150 can be used to remove leader peptides from diverse unmodified and modified peptides and hence is a wonderful tool for RiPP bioengineering and genome mining (leader peptide removal has been a real hurdle since no commercial enzymes cleave at double Gly motifs and engineering a commercial protease cleavage site into the peptides usually results in the PTM enzymes no longer functioning).

In the former Figure 1A, we had intended to show that many PCATs act on peptides that can carry a plethora of different posttranslational modifications in their “cargo” (posttranslationally modified core peptides in the case of RiPPs). We chose thioether rings as an example, but we agree that this is not particularly well explained or justified, nor easily picked up unless the reader perhaps works in this area. We could not think of a better way to illustrate what types of modifications are relevant here without knowing what RiPP the Lah cluster makes (which would be a completely different project), and hence we elected to delete the former Figure 1A.

3) Figure 1B. While I appreciate the high sequence similarity between the leader peptide candidates, I have a more difficult time appreciating the sequence diversity on the core peptide. An additional panel highlighting the diversity of amino acids accepted after the double Gly motif would be helpful.

We thank the reviewer for the suggestion. Rather than making a new panel, we added a sequence conservation logo to the figure for the core peptide, which illustrates the very low level of conservation in the core peptide.

4) Figure 1C. I am confused by Figure 1C since I could not find a reference in the main text until the last paragraph of the subsection “Identification of a substrate tolerant protease”. The first reference in "Results" refers to Figure 2. It is not self-explanatory why mass spec data showing proteolytic activity is shown here since leader peptide removal and inhibitor design receive a dedicated figure with Figure 3.

We now have a dedicated figure for leader peptide removal from the cognate substrates from the *lah* cluster (Figure 2). In our revised Figure 4 (derived from the former Figure 3), we show data with non-cognate peptides to illustrate the tolerance of the LahT150 protease domain with respect to both leader peptide and core peptide (unmodified and even non-cognate modified peptide).

5) Figure 4A and D. In both cases it is unclear to me why these panels exist. The main text takes no note of them which is particularly unfortunate in the case of Figure 4D. The main text notes an apparent narrowing of the substrate binding pocket towards the double Gly motif and the figure suggests this to be accompanied by an increase in hydrophobicity.

We have now added sections into the main text that better integrate Figure 4A and D into the text.